# AlphaFold predictions of fold-switched conformations are driven by structure memorization

Devlina Chakravarty[1], Joseph W. Schafer[1], Ethan A. Chen[1], Joseph F. Thole [1,2], Leslie A. Ronish[1,2], Myeongsang Lee[1] & Lauren L. Porter [1,2] ✉

Recent work suggests that AlphaFold (AF)–a deep learning-based model that can accurately infer protein structure from sequence–may discern important features of folded protein energy landscapes, defined by the diversity and frequency of different conformations in the folded state. Here, we test the limits of its predictive power on fold-switching proteins, which assume two structures with regions of distinct secondary and/or tertiary structure. We find that (1) AF is a weak predictor of fold switching and (2) some of its successes result from memorization of training-set structures rather than learned protein energetics. Combining >280,000 models from several implementations of AF2 and AF3, a 35% success rate was achieved for fold switchers likely in AF's training sets. AF2's confidence metrics selected against models consistent with experimentally determined fold-switching structures and failed to discriminate between low and high energy conformations. Further, AF captured only one out of seven experimentally confirmed fold switchers outside of its training sets despite extensive sampling of an additional ~280,000 models. Several observations indicate that AF2 has memorized structural information during training, and AF3 misassigns coevolutionary restraints. These limitations constrain the scope of successful predictions, highlighting the need for physically based methods that readily predict multiple protein conformations.

Deep learning-based algorithms have made it possible to predict protein structure from amino acid sequence, sometimes with impressively high accuracy. The most successful of these algorithms, AlphaFold2 (AF2)[1], has inspired numerous approaches to predict and design other important structural features of proteins. These features include protein-protein interaction sites[2], conditionally folding regions of intrinsically disordered proteins[3], and structures of previously uncharacterized protein folds from metagenomic sequences[4]. Furthermore, AF's successor, AlphaFold3 (AF3)[5], enables modeling of interactions between proteins and other biomolecules with impressive success.

The many successes of AF (AF2 and AF3) suggest that it may also predict subtle-yet-important protein properties previously revealed only through sophisticated techniques. These properties include conformational ensembles and functionally important alternative conformations[6]. Consistent and accurate predictions of these properties would suggest that AF may do more than simply associate protein sequence with structure through sophisticated pattern recognition[7]. Rather, it may leverage learned folding physics to accurately approximate folded protein energy landscapes[8]. These landscapes are defined by the diversity and frequency of protein conformations in the folded state. Supporting this possibility,

[1]National Center for Biotechnology Information, National Library of Medicine, National Institutes of Health, Bethesda, MD 20894, USA. [2]Biochemistry and Biophysics Center, National Heart, Lung, and Blood Institute, National Institutes of Health, Bethesda, MD 20892, USA. ✉e-mail: porterll@nih.gov

AlphaFold2 has successfully predicted alternatively folded states in over a dozen protein families[6,9,10].

Yet despite AF's impressive accuracy and broad success, several uncertainties remain about how much it has learned about protein energy landscapes. These uncertainties relate to the two major tasks on which protein structure prediction relies: adequate sampling and accurate scoring. First, sampling refers to AF's ability to generate distinct experimentally consistent conformations of the same protein. As a deep learning algorithm, AF relies on a large training set of solved and predicted structures, their amino acid sequences, and multiple sequence alignments (MSAs) containing evolutionary information used to infer structure[1]. Though AF2's training set is not publicly available, the training set of OpenFold[11], which uses the same software architecture and predicts protein structure with similar accuracy, contains >130,000 unique protein chains[12]. Based on its published methods, AF3's training set likely contained a similarly large number of experimentally determined structures[5]. Compared to these large training sets, the number of proteins with multiple distinct experimentally determined conformations is small[13]. Furthermore, AF's ability to sample multiple experimentally determined conformations has been tested on only a handful of examples[6,9,10]. Thus, it is unknown how well AF would sample multiple protein conformations more broadly[13,14]. Second, scoring refers to AF's ability to distinguish between good and poor predictions. Currently, AF2 assigns good and poor scores to its predictions of single protein conformations very reliably[8]; the overall quality of AF3's scoring has not yet been assessed. To our knowledge, however, no studies have systematically assessed how accurately AF2 scores alternative protein conformations, though a recent study reports that its confidence metrics are not reliable for a handful of multi-conformational proteins[15].

In previous work, we hypothesized that AF2 may be using sophisticated pattern recognition to search for the most probable conformer rather than learned energetics to model a protein's structural ensemble[16]. This work was a straightforward implementation of an older version of AF2 (2.0) with no enhanced sampling techniques. Since then, several recently developed enhanced sampling techniques have challenged our hypothesis, proposing instead that AF2 couples coevolution with a learned energy function to predict alternative conformations[10,15,17]. These methods were tested on a handful of targets (6-16/study), however, leaving open the questions of (1) how well they generalize across a class of proteins and (2) what systematic benchmarking results may reveal about AF2's overall ability to predict alternative protein conformations. Furthermore, AF3 has also just been released as a webserver[5] and has not yet been tested on fold-switching proteins.

In this work, we investigate whether AF-based predictions are driven by pattern recognition or a learned energy function by systematically assessing AF's ability to sample and score both experimentally determined conformations of 92 fold-switching proteins[18]. This emerging class of proteins has been evolutionarily selected to assume two distinctly folded states[19], presumably for functionally important reasons[20]. Though the energy landscapes of these fold-switching proteins are populated by many more conformations than their two distinct experimentally determined conformations[21–23], we use these two fold-switching conformations as a minimalist approximation of a folded protein energy landscape. After all, the energetic metric of choice for protein structure is Gibbs free energy, which directly relates stability to observational frequency. Importantly, the structures in our dataset were carefully curated to include functional explanations and trigger for both conformations, eliminating false positives resulting from crystal packing artifacts[18]. This curation has stood the test of time: recent work indicates that evolution has selected for both conformations of many proteins in this dataset[19]. Thus, we hypothesize that these experimentally observed conformations of fold switchers are likely major constituents of their energy landscapes. This hypothesis is supported by multiple experimental and computational observations[21,22,24]. Thus, we posit that if AF has truly learned protein energetics, it should consistently and accurately predict both experimentally observed conformations of fold-switching proteins. If not, then pattern recognition is the likely driver of some AF-based predictions.

Here, we present an up-to-date assessment of AF's ability to predict fold-switching proteins. Previously, we showed that AF2.2.0 is systematically biased to predict one conformation of fold switchers while missing the other[16]. Since then, AF2.3.1 has been released: this version now makes accurate predictions of oligomeric assemblies and protein-protein interactions[25]. Because at least one conformation of most fold switchers forms an oligomer or interacts with another protein[18], we aimed to assess AF2.3.1's ability to predict fold switching when information about oligomeric state and/or binding partner is provided. Both conformations of all 92 fold switchers were deposited in the Protein Data Bank[26] (PDB) before AF2.3.1 was trained, and all pairs of conformations were in the OpenFold training set[12], suggesting that they are likely in AF2's training set as well. Based on AF3's reported methods, 78/92 (85%) of these fold switchers are likely in its training set as well (all pairs without an NMR structure). Thus, we tested AF3 on all fold-switching pairs and included as many relevant interacting biomolecules in our modeling as possible. Finally, two methods for predicting alternative protein conformations or protein ensembles with AF2 have recently been proposed[15,17]. Thus, we tested the performance of these methods on the same set of 92 fold switchers, generating >280,000 predictions in all. Upon assessing these predictions, we found that all AF2-based methods and AF3 predict fold-switching proteins likely in its training set with modest success (32/92). Further, AF2's confidence metrics select against alternatively folded protein conformations and cannot discriminate between low and high energy conformations of fold-switching proteins. Because AF's predictions are most useful for proteins without experimentally determined structures, we also tested AF2 and AF3 on a set of seven fold-switching proteins whose structures were deposited in the PDB or confirmed by other experimental methods after they were trained, generating ~280,000 additional predictions. They failed to predict the alternative folds of 6/7 fold switchers.

Since these results demonstrate that AF has not fully learned protein folding energetics, we sought explanations for why. We found that AF2's predictive success for some fold-switching proteins results from "memorization" of structures in its training set. This memorization can be so strong that AF2 uses it to inform predictions instead of coevolutionary information detected by its Evoformer. Furthermore, AF3 generated an incorrect prediction of human lymphotactin by misassigning the evolutionary restraints it detected. These limitations explain AF's frequent failure to predict fold switchers outside of its training set and constrain AF's ability to predict alternative conformations yet to be discovered.

## Results

### AF samples both conformations of 35% of fold switchers likely in its training set

AF's ability to sample two folds assumed by single sequences was tested on 92 pairs of experimentally determined fold switchers. To our knowledge, these 92 pairs (Supplementary Data 1) include fold switchers from from many diverse fold families and source organisms[16]. These structural pairs are likely in AF2.3.1's training set because they were all deposited in the PDB before 2022, and all of them were in the training set of OpenFold[12,27], an AI-based model with the same architecture and performance as AF2. Most of them (85%) are also likely in AF3's training set, based on the published methods[5]. All protein pairs have identical or nearly identical sequences and regions of distinct secondary and tertiary structure. AF predictions are defined as successful when they accurately capture both experimentally

determined conformations, called Fold1 and Fold2. Prediction accuracy is assessed by calculating the TM-score[28] between each AF prediction and both experimentally determined conformations. TM-scores quantify the similarity of topology and connections between secondary structure elements[29], a reliable metric since fold-switching proteins are identified by secondary structure differences[18]. Because whole-protein TM-scores often overestimate the prediction accuracies of fold-switching regions, we assessed predictions using TM-scores of fold-switching regions only (Supplementary Fig. 1). Higher TM-scores indicate predictions closer to experimentally determined conformations. We ordered each pair of fold switchers so that Fold1 corresponds to the target conformation most frequently predicted by AF2, and Fold2 corresponds to the less frequently predicted target conformation (Methods: *Defining Fold1 and Fold2*). To augment this TM-score-based assessment, we also performed root-mean-square-deviation (RMSD) calculations of fold-switching regions and found similar results (Supplementary Fig. 2).

First, four different AF2.3.1 modes and AF3 were tested on each fold-switching sequence: with templates, without templates, multimer model on single chains, and multimer model on protein complexes (Supplementary Data 2). AF2.3.1's performance increased slightly above AF2.0's (Fig. 1a), capturing 11/92 fold switchers (combining results both with and without templates) rather than 8/92[16]. Furthermore, AF2_multimer successfully predicted both conformations of 12/92 fold switchers. Surprisingly, AF3 underperformed relative to AF2, capturing both conformations of 7/92 fold switchers in total. Since AF3 was updated to model interactions between proteins and other biomolecules, we included as many binding partners as possible in the modeling: DNA, RNA, ions, and other ligands (Supplementary Data 3). Although the AF3 webserver is currently limited to a subset of biomolecules, 70% of the interactions in our dataset could be fully modeled with the ligands available (115/165).

Although fold switching is often triggered by interactions with other proteins or biological molecules[18], supplying this information to the Multimer model and AF3 yielded only nine unique fold-switch predictions, seven of which were predicted using single chains by other AF2.3.1 methods (Supplementary Data 2). Both the TM-score and RMSD-based assessments demonstrated that running AF2.3.1 with default inputs and parameters and default AF3 with appropriate interacting biomolecules infrequently produce successful fold switch predictions: 21% combined.

We then tested whether AF2-based enhanced sampling approaches can predict more fold switchers than AF runs with standard inputs. Recently, two such approaches have been proposed to predict alternative conformations of proteins including fold switchers. The first, SPEACH_AF[17], masks coevolutionary information in AF2's input MSA by mutating selected columns to alanine in silico. Masking this information is expected to allow AF2 to identify coevolutionary signals in the MSA corresponding to alternative protein conformations, allowing it to sample a more diverse conformational ensemble. SPEACH_AF was tested on 16 different proteins and generated alternative conformations for almost all of them. Though none of these proteins were fold switchers, SPEACH_AF's potential to predict fold switching was proposed[17]. The second approach, AF-cluster[15], clusters sequences from a deep MSA by similarity and runs AF2 on individual clusters. This approach is based on the hypothesis that different MSA subsets may contain coevolutionary information distinct from deep MSAs, allowing AF2 to predict alternative protein conformations, though recent work suggests that AF-cluster may infer alternative conformations from its PDB training rather than coevolutionary inference[30], limiting its robustness. Regardless, AF-cluster was tested on six families of fold-switching proteins and successfully predicted both conformations in three families[15].

To gauge how frequently SPEACH_AF and AF-cluster predict fold switching, we tested both approaches extensively on the set of 92 fold

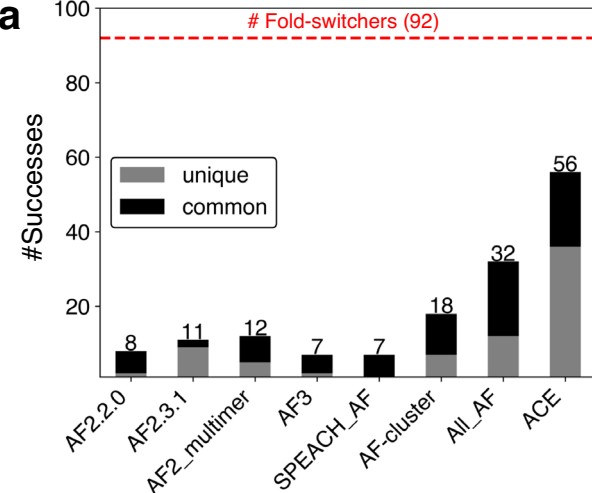

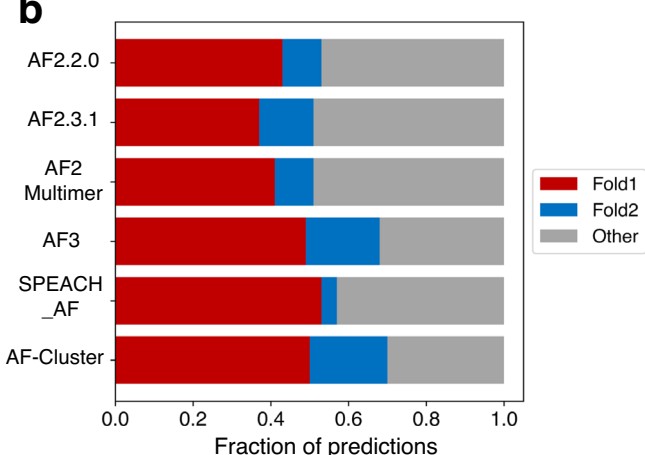

**Fig. 1 | AF predicts fold switching with modest success. a** Numbers of successful fold-switch predictions for each AF2 method and AF3 compared with coevolutionary information found for both folds (ACE) and the total number of possible successes (dotted red line). All_AF2 combines all unique successful predictions from all AF2-based methods: >282,000 predictions. Predictions successfully made by more than one AF method are black; predictions unique to each method are gray. **b** Fraction of predicted structures that match experimentally determined conformations for all methods. Fold1 is the conformation most frequently sampled by AF2.3.1, Fold2 is the less frequently sampled (or unsampled) conformation. Conformations designated as Other are inconsistent with both experimentally determined structures. Source data are provided as a Source Data file.

switchers tested previously, generating >77,000 structures with SPEACH_AF and >200,000 structures with AF-cluster (Supplementary Data 2). Both methods missed fold switching in most cases (Fig. 1a): 92% for SPEACH_AF (7/92 successes) and 80% for AF-cluster (18/92 successes).

As mentioned previously, both SPEACH_AF and AF-cluster postulate that AF2 can predict alternative protein conformations when sufficient coevolutionary information is provided. A recent computational approach called Alternative Contact Enhancement (ACE) identified coevolutionary information unique to both folds of 56 fold-switching proteins, confirming that MSAs often contain structural information unique to both conformations[19]. Nevertheless, after combining all correctly predicted fold switch pairs from 282,000 predicted structures (Fig. 1b), AlphaFold2 misses this information in 35/56 cases. Thus, current enhanced sampling approaches typically do not enable AF2 to consistently detect the dual-fold coevolutionary information present in many MSAs of fold-switching proteins.

## AF2 confidence metrics select against alternative conformations of fold switchers

Though AF2 often produces structural models with remarkably high accuracy[1], its accuracy is reduced for fold-switching proteins when shallow MSA subsampling is used. We quantified the frequency of inaccurate predictions relative to correct predictions of Fold1 and Fold2 generated by all methods (Fig. 1b). In all cases, 30–49% of predictions did not correspond well to either experimentally determined structure.

To see if AF2 could distinguish between good and inaccurate predictions, the relationship between prediction quality and AF2's confidence metrics was assessed. AF2 estimates prediction quality with two confidence metrics: the per residue predicted Local Difference Distance Test (plDDT) and predicted template modeling (pTM) scores. We sought to determine whether either or both metrics discriminate between the good and poor fold-switch predictions generated by AlphaFold2 and AF-cluster. AF-cluster was selected because it predicted substantially more fold switchers than SPEACH_AF (18 rather than 7), generated fewer inaccurate predictions overall (~30% rather than 43%), and enabled a larger set of diverse predictions to be made.

Neither of AF2's confidence metrics successfully discriminated between good and inaccurate fold-switch predictions (Fig. 2a, Supplementary Figs. 3–5). Rather, both plDDT and pTM scores assigned lower confidences to diverse correctly predicted conformers and higher confidences to predictions that have not been observed experimentally. Thirty percent of all AF-cluster structures did not match experimentally determined structures of Fold1 or Fold2, making it the most accurate of all AF-based methods (Fig. 1b). However, of its highest ranked structures, the proportion of predictions inconsistent with experiment increased to nearly 70% (Fig. 2a, Supplementary Data 4, 5). A similar trend was observed for AF2.3.1 runs with standard settings (Supplementary Figs. 4–5). Interestingly, upon dividing targets into "Easy" and "Complex" based on the type and amount of conformational change, "Complex" targets were better represented in the "Top10" and "All" categories than "Easy" at all quality levels (Supplementary Data 6).

These results strongly indicate that AF2's confidence metrics select against experimentally consistent predictions of fold switchers, especially Fold2, in favor of experimentally inconsistent predictions. For instance, while AF-cluster correctly predicted 18/92 Fold2 conformations overall, only 7/92 were identified amongst high quality predictions ($p < 8.1 \times 10^{-4}$, one-sided binomial test). Further, significantly fewer correctly predicted conformations (either Fold1 or Fold2) were identified amongst high-quality models (37) than amongst all (53, $p < 6.6 \times 10^{-4}$, one-sided binomial test).

Some of the experimentally unobserved conformations predicted by AF2 have been proposed to correspond to folding intermediates[6]. To the best of our knowledge, there is no experimental evidence supporting this claim for fold-switching proteins. In fact, a recently characterized folding intermediate of the transcriptional regulator RfaH suggests the opposite[22]. AF2-multimer predicted a hybrid α-helical/β-sheet fold with high confidence for its fold-switching C-terminal domain (Supplementary Fig. 6). This prediction is not consistent with experiment: most notably, the N-terminal portion of the AF2 prediction folds into a β-hairpin, while the experimentally observed intermediate has helical propensities in that region[22]. Thus, high confidence AF2 predictions that differ from experimentally determined structures do not necessarily correspond to folding intermediates, consistent with previous observations[31].

To further address if AF2-based enhanced sampling methods can predict folding intermediates, we compared models of essential mannosyltransferase PimA from *M. tuberculosis* with structures from accelerated molecular dynamics (MD) simulations consistent with [19]F NMR experiments. These experiments revealed four functionally relevant states of PimA that coexist in dynamic equilibria[21]: two stable

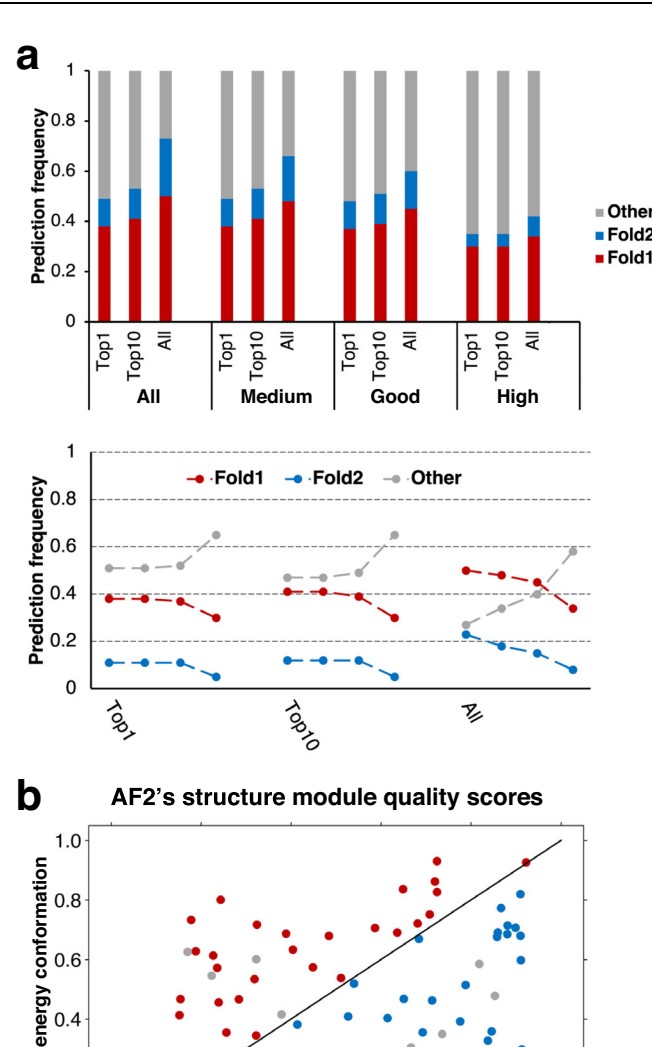

**Fig. 2 | AF2 confidence metrics select against alternative conformations and do not predict the most energetically favorable fold-switch conformations. a** Barplot representation of prediction success in Top1, Top10 and All fold-switch predictions indicate that more experimentally unobserved conformations are selected as prediction confidence increases. These trends are apparent in trendline plots showing the change in fraction of predictions as a function of prediction confidence. The leftmost 3 trendlines are from All predictions, the middle/rightmost are from Top10/Top1 most confident for each of 92 fold switchers. For each column of trendlines, the leftmost dot represents all conformations (not weighted by confidence), the next is predictions with medium confidence, then good confidence, and finally high confidence. Confidences are determined by ≥70% (medium), 80% (good), 90% (high) of residues with Cα plDDT scores ≥70. **b** AF2's structure module predicts the lower energy conformations of fold switchers with better accuracy and higher confidence than higher energy conformations 50% of the time, equal to random chance. Blue dots represent correctly predicted ground state conformers with higher confidence; red dots represent correctly predicted excited state conformers with higher confidence than low energy, and gray dots have been observed to sample both folds at roughly equal proportions at equilibrium. Axes represent TM-scores of both conformations relative to experiment. Source data are provided as a Source Data file.

fold-switched states and two intermediates. Since only AF-cluster successfully predicted the both stable fold-switched states of PimA, we searched among its ~1400 models for structures resembling the two intermediates (Methods: *PimA Intermediates*). None were found. Among the models, 47% resembled Fold1 (active-compact state of PimA), 0.2% resembled Fold2 (inactive-compact or apo state), and the remaining ~53% of predictions did not resemble any of the four states, though many of these predictions (50%/53%) had low confidence (average plDDT <70).

AF2's inability to discriminate between good and poor predictions of fold switchers suggests that its confidence metrics may have broader limitations. To further assess this possibility, we used AF2's structure module to energetically rank fold-switching protein pairs (Methods: *AF2Rank*). This approach correctly selected experimentally consistent structures among diverse models of 283 proteins[8]. Here, it correctly selected the ground state conformations of fold-switching proteins 50% of the time (Fig. 2b). In other words, the selective power of AF2's structure module amounted to random guessing for fold-switching proteins. It may seem reasonable to hypothesize that this selective failure arises in cases where the ground states of fold switchers are oligomeric and the excited states are monomeric. This may not be the case, however, because AF2 predicts the folds of ground state oligomeric structures, such as KaiB, with the monomer model[15]. Furthermore, including oligomeric states and binding partners in the multimer model did not produce any unique fold-switch predictions (Supplementary Data 2); instead, all alternative conformations were predicted from monomeric sequences without the need for additional information about oligomeric state or binding partner. Thus, AF2 does not seem to require additional information about oligomeric state or protein binding partner to predict conformations of proteins in oligomeric assemblies or complexes. Providing additional biomolecular information to AF3 did not appreciably increase its predictive success either: out of its 7 successes, only 2 were not predicted by AF2.

## AF rarely predicts fold switchers outside of its training set

AF's modest success in sampling the conformations of fold switchers likely within its training set raises the question of how well it can predict fold switching of sequences without. After all, AF is most valuable when used to infer structural properties of uncharacterized proteins, such as conditionally folding regions of IDPs[3] and yet-to-be-discovered folds[4]. Thus, we identified seven fold switchers with sequences outside of AF's training sets and divided them into two categories: distant homologs of a known fold switcher and recently discovered fold switchers. The alternative conformations of all seven fold switchers were either (1) determined after AF2.3.1's and AF3.0's last training or (2) inferred by other experimental methods without depositing the alternative structure in the PDB.

First, we assessed AF's ability to predict fold switching of five distant homologs of the known fold-switching protein *Escherichia coli* RfaH[32], a bacterial transcription factor whose C-terminal domain reversibly switches from an all α-helical ground state to an all β-sheet excited state upon binding RNA polymerase and a specific DNA sequence called *ops*[33]. Both conformations of *E. coli* RfaH have been determined experimentally[34,35]. Previous work provided circular dichroism (CD) and nuclear magnetic resonance (NMR) evidence for switching in all five of these sequence-diverse RfaH homologs[32], all with sequences <35% identical to one another's and to *E. coli* RfaH's. As a control, AF's ability to predict single folding was assessed in five additional experimentally characterized single-folding RfaH homologs whose CTDs were found to assume the β-sheet fold only (Supplementary Table 1).

Although AF2, AF3, and AF-cluster correctly predict that *E. coli* RfaH—likely in their training sets—switches folds, none of them reliably predicted fold switching in the experimentally confirmed variants not deposited in the PDB. Specifically, AF2.3.1 and AF3.0 predicted a helical CTD in 1/5 cases with moderate confidence (Supplementary Fig. 7). In the other four cases, they predicted the β-sheet conformation only, as they did correctly for all single-folding controls. To extensively search for fold switching with AF-cluster, we generated 50 models per input MSA with 10 seeds for a total of 140,050 predictions of 10 proteins (Supplementary Data 7) both with and without dropout (>280,000 structures total), plus 5 models per input MSA with 2 seeds using both ColabFold1.3 and 1.5. Combining all predictions, AF-cluster predicted both folds for 4/5 conformations and only well-folded β-sheet conformers in the remaining case (Supplementary Fig. 8). However, all helical conformations were predicted with low confidence (average plDDT ≤50), indicating that AF2.3.1 can generate more confident helical CTD predictions than AF-cluster. This finding is consistent with the original AF2 paper's observation that MSAs with ≥32 sequences are needed for reliable predictions[1]; AF-cluster-generated MSAs often have ≤10 sequences. Importantly, AF-cluster predicted low-confidence helical conformations in two single-folding RfaH homologs with CTDs experimentally confirmed to assume β-sheet folds rather than α-helices (Supplementary Fig. 8). NMR evidence from a previous study strongly suggests that the *Candidatus Kryptonium thompsoni* variant assumes the β-sheet conformation only[32]. Furthermore, the CD spectrum of the *T. diversiorginum* variant also suggests that it assumes a ground state β-sheet structure consistent with previously characterized RfaH variants whose CTDs do not assume helical conformations (Supplementary Fig. 9). Together, these results demonstrate that neither AF2, nor AF3, nor AF-cluster reliably predict fold switching of distant RfaH homologs, and AF-cluster predictions do not reliably distinguish between fold-switching and single-folding RfaH variants.

Structures of the two remaining prediction targets were deposited into the PDB in 2023, after AF2.3.1 and AF3 were trained. Fold switching of Sa1–a 95 amino acid protein that reversibly interconverts between a 3-α-helix bundle and an α/β plait fold in response to temperature–was demonstrated by NMR spectroscopy[36]. We also included the structure of BCCIPα, a human protein whose sequence is 80% identical to its homolog BCCIPβ. Although BCCIPα has not been shown to switch folds, it assumes a structure completely different from BCCIPβ and has a different binding partner than its homolog[37]. Previous work has shown that when run with default parameters, AlphaFold2 fails to predict the unique structure of BCCIPα, whose most similar PDB analog differs by 9.9Å[37]. Thus, we included BCCIPα because (1) we wanted to see if AF-cluster or AF3 could produce its unique structure and (2) although BCCIPα might not switch folds, it tests AF2's limits in predicting protein folds outside of its training set.

AF2.3.1, AF3, and AF-cluster missed fold switching completely for both Sa1 and BCCIPα (Fig. 3). Specifically, 98.8% (2525/2555) of the Sa1 predictions assumed the α/β plait fold, and 54% (2022/3755) of the BCCIPα predictions assumed the structure of its PDB homolog, BCCIPβ. By contrast, AF2.3.1, AF3, and AF-cluster failed to predict both the 3-α-helix bundle conformation of Sa1 and the experimentally determined conformation of BCCIPα. BCCIPα's structure was solved in complex with another protein[37]. Nevertheless, running AF2.3.1's Multimer model and AF3 with BCCIPα's binding partner still yielded the BCCIPβ structure (Supplementary Fig. 10). Because its apo structure has not yet been determined, it is possible that apo BCCIPα assumes the same structure as BCCIPβ, in which case AF2.3.1, AF3, and AF-cluster fail to predict its alternative conformation. It is also possible that apo BCCIPα assumes the same structure in its apo and bound forms, in which case AF2.3.1, AF3, and AF-cluster fail to predict its structure altogether[38]. These results cast doubt on the AF's reliability and consistency in predicting the alternative conformations of fold switchers outside of its training set.

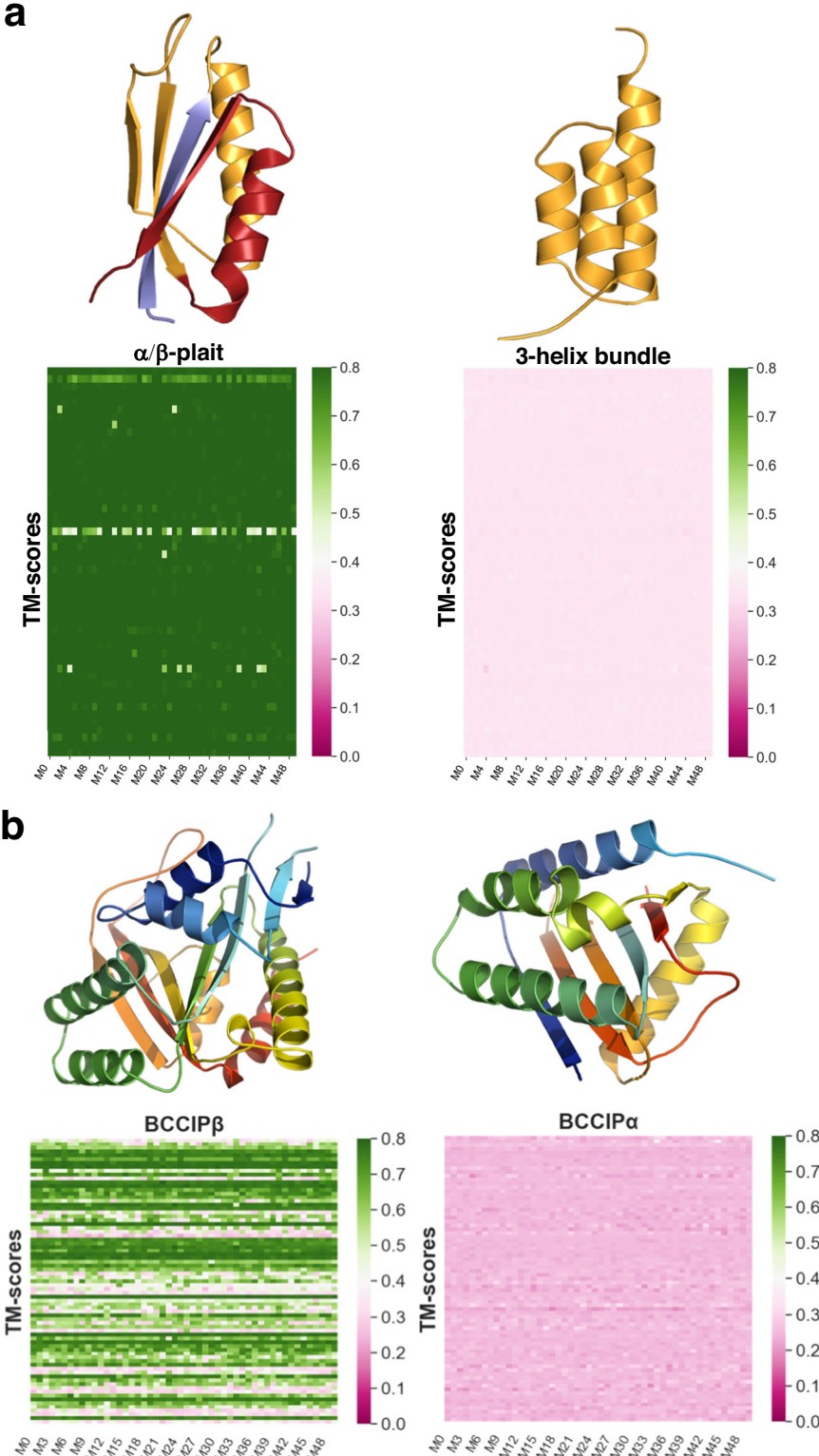

**Fig. 3 | AF2 fails to predict fold switching of two protein structures outside of its training set. a** Sa1 is a designed protein that switches reversibly between α/β-plait (PDBID:8e6y, Fold1) and 3α helix (PDBID: 2fs1, Fold2) folds triggered by temperature changes. Cartoon representations of Fold1 are colored blue for N-terminal residues (1 to 10), orange for the fold-switching residues (11 to 66 aligning with the amino acid sequence in Fold2, also in orange) and C-terminal residues (67 to 95) are red. Heatmaps of 50 predictions (M0 to M49) for each of 51 sequence clusters showing the similarity (TM-scores) to Fold1(left panel) and Fold2 (right) are presented below the cartoon representations of the two states. AF-cluster consistently predicts Fold1 but misses Fold2. **b** BCCIPβ and BCCIPα are

human protein isoforms with 80% sequence identity that adopt distinct folds. (13 Å RMSD). AF-cluster was run on BCCIPα's sequence. In the right panel, a cartoon representation of BCCIPα (colored blue to red from N-terminus to C-terminus) is shown with the heatmap of TM-scores of 50 predictions (model numbers M0 to M49) for each of 75 sequence clusters compared to the fold adopted by the α isoform (PDBID:8exf, chain B). In the left panel, the BCCIPβ experimental structure (PDBID:7kys) is shown with the heatmap of TM-scores compared to the fold adopted by the β isoform. AF-cluster frequently predicts the structure of the β-isoform but misses the experimentally consistent α-isoform structure.

### AF2 predictions are not always consistent with coevolutionary restraints and are better explained by memorization of training set structures

Why does AF2 fail to predict alternative conformations outside of its training set? Two dominant explanations have been proposed. The first is insufficient information. AF2 has been proposed to use MSA-derived restraints as a starting point to minimize the energies of structures, much like NMR structure determination[8]. If AF2 works this way, its failure to predict a given conformation would result from improper restraints, i.e. the input MSA did not supply the information needed to specify the fold of interest. The second is structure "memorization". In this case, AF2 does not always rely on coevolutionary restraints because it has "seen" certain folds during training and stored relevant structural information in its weights[13], allowing it to associate learned structures with related sequences. The distinction between these two explanations is important. If AF2 predicts structures by energy minimizing structural restraints from MSAs, it can, in principle, predict any yet-to-be-discovered fold from its sequence given proper MSA input. By contrast, if AF2 relies on its training set to predict certain structures, it may be unable to correctly associate some sequences with their corresponding structures. This may explain its failure to predict the correct structure of BCCIPα and most RfaH variants, for instance. It also suggests that both structures of Sa1 may be predicted if AF2 can be steered to associate its sequence with the homologous 3-α-helical bundle conformation in its training set.

We applied our knowledge of AF2's architecture to assess how it predicts alternative conformations (Supplementary Fig. 11). AF2 combines two modules to predict protein structure. The first is the Evoformer, which extracts evolutionary couplings from input MSAs

and stores them as a pair representation, a tensor of real numbers used to predict distances between each amino acid pair in a protein chain. The pair representation and the target sequence are then passed to the Structure module, which maps these inputs to a three-dimensional structure. This predicted structure, along with the pair representation can be passed back into the AF2 network for further rounds of refinement, a process called recycling. Thus, before recycling, the pair representation is informed by the input MSA only. After recycling, the pair representation is updated with information both from the MSA and the protein model generated by the Structure module (Supplementary Fig. 11). Consequently, coevolutionary information that AF2 derives from an input MSA can be assessed most reliably at 0 recycles, since the MSA does not exclusively supply the information used to inform the pairwise representation after recycling.

Leveraging this knowledge, we observed that ColabFold[39] (CF)–an efficient-yet-accurate implementation of AF2–predicts structures of *E. coli* RfaH inconsistent with the restraints it infers from MSAs at each recycling step (Fig. 4). Specifically, by leveraging coevolutionary information from its input MSA at 0 recycles, CF predicts the active conformation of RfaH with a fully β-sheet C-terminal domain (CTD). Interestingly, at subsequent recycling steps, its CTD becomes increasingly helical, resembling the autoinhibited state. Since CF updates the input MSA at the beginning of each recycling step, this structural change could arise from updated MSA-based coevolutionary information updating the pair representation. This was not the case, however, when we inputted each updated MSA into CF with 0 recycles. Instead, CF predicted structures with fully β-sheet CTDs from all MSAs (Fig. 4). Thus, AF2's MSA-derived pairwise restraints are inconsistent

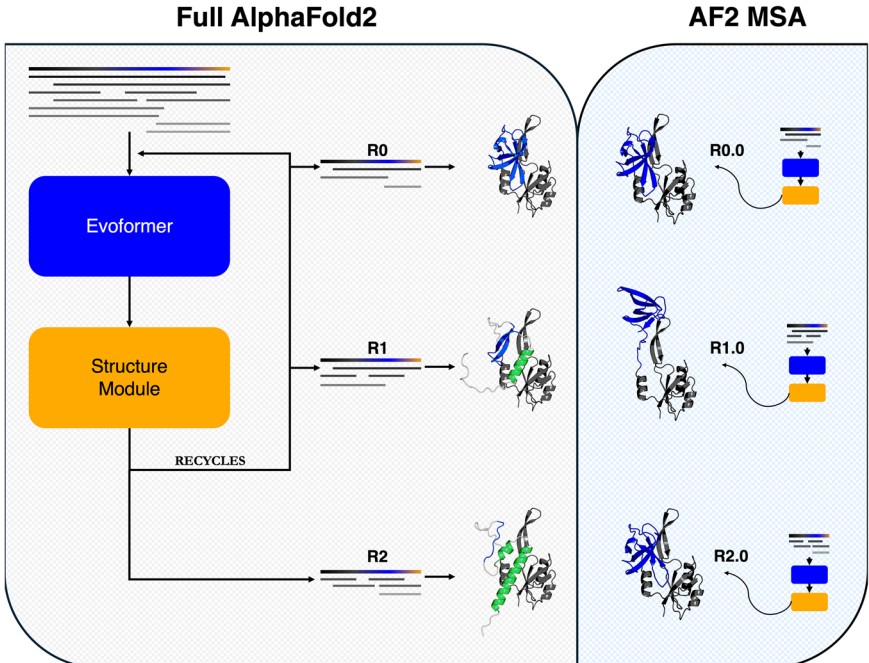

**Fig. 4 | AF2 structure predictions can be inconsistent with structural restraints from Evoformer.** Although the full AF2 model predicts the autoinhibited form of RfaH (green helical structure, left panel) after 2 recycles (R2), the evolutionary restraints from Evoformer correspond to its active β-sheet form (blue β-sheet structures, right panel and Fig. S12) from each MSA inputted into the full AF2 model (left panel). The initial input MSA is depicted in the top lefthand corner with target sequence bold and colored black, blue, and yellow. Randomly subsampled MSAs inputted at each recycle are depicted in both panels, with identical MSAs being inputted at R0,1,2 and MSA_R0.0, MSA_R1.0, MSA_R2.0, respectively. The right and left panels differ by how AF2 makes predictions. In the right panel, restraints from

input MSAs should inform the predictions because the input MSA is passed through AF2 (Evoformer and Structure Module) only once (0 recycles); this also applies to the R0 (0 recycles) step in the left panel. All structures based on these MSA restraints output structures with β-sheet CTDs (blue). The recycling steps in the left panel (R1 and R2) differ because they update the prediction with both previous MSA restraints and the previously predicted structures from the Structure Module. In these cases, the CTD becomes increasingly helical (green regions), indicating that the prediction changes during the recycling process.. Right and left panels are shaded to represent what information drives predictions: beige (recycling process, left) and light blue (Evoformer, right).

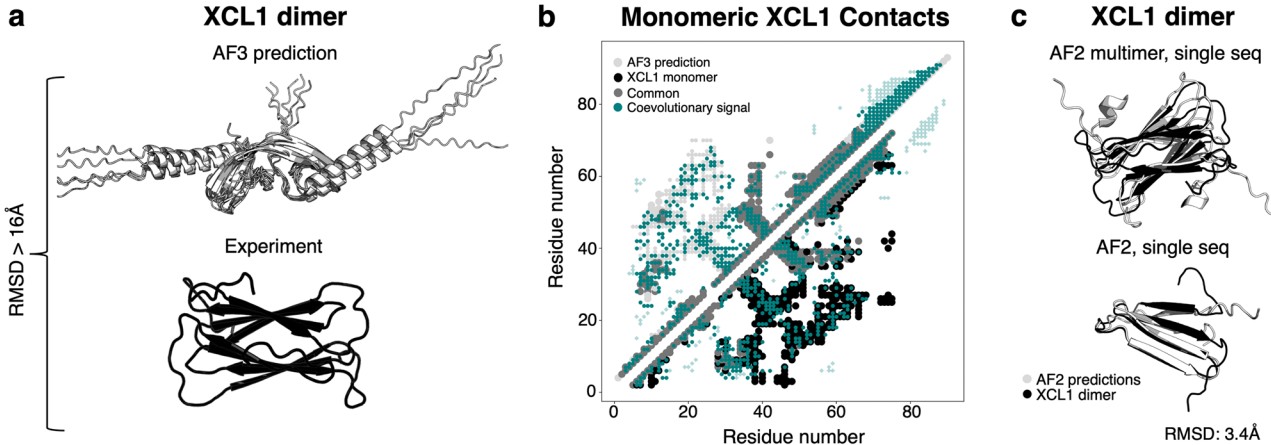

**Fig. 5 | AF predicts the dimeric form of XCL1 through structure memorization rather than coevolutionary inference. a** Though AF3 was given appropriate stoichiometry and environmental conditions to predict the lymphotactin dimer, its prediction did not match experiment. **b** The coevolutionary patterns of AF3's predicted XCL1 dimer match those of its monomeric conformation almost exactly. Contact maps generated from the AF3 prediction and experimentally determined monomeric conformation (2hdm). Upper diagonal corresponds to contacts unique to the AF3 prediction (light gray smaller dots correspond to intermolecular contacts; larger to intramolecular), lower diagonal corresponds to contacts unique to the experimentally determined monomeric conformation (black); common contacts medium gray; coevolutionary information inferred from MSA using ACE, (teal). **c** Both AF2 multimer and AF2 predict the correct XCL1 dimer structure from single sequences and 0 recycles, suggesting that they memorized its structure during training.

with those from its recycled structure predictions (Supplementary Fig. 12), indicating that the autoinhibited prediction of RfaH likely arises from something other than evolutionary restraints inferred from its input MSA.

Since the coevolutionary patterns that CF recognizes are inconsistent with its recycled prediction of autoinhibited RfaH, we sought to identify what drives this prediction. Previous work has suggested that predictions may sometimes be informed by structures "memorized" during training[13]. This seemed like a reasonable explanation for the inconsistencies we observed between evolutionary couplings and predicted structures (Supplementary Fig. 12).

To test the possibility that the autoinhibited form of RfaH's CTD may have been memorized during training, we inputted the single sequence of RfaH's CTD into CF and examined its predictions after 0 recycles. This assessment focuses on what may have been "memorized" during PDB training since (1) the Evoformer cannot determine amino acid covariances from a single sequence and (2) 0 recycles affords an initial structural guess only from the target sequence, whereas recycling would allow deeper exploration of the AF2 network and may not suggest memorization. Out of the 25 RfaH CTD models generated, CF predicted that it forms a helical bundle 100% of the time (Supplementary Fig. 13a). These predictions contradict experimental observation: expressed in isolation, the RfaH CTD folds into a β-sheet structure, not a helical bundle[40]. This result again demonstrates AF2's limited learning of protein energy landscapes. It also indicates that AF2 has likely memorized RfaH's helical bundle conformation during training since predictions consistently resemble the helical structure likely in AF2's training set. To probe for other cases of putative structural memorization, we performed single-sequence predictions on other fold-switching sequences and identified resulting models consistently resembling their corresponding PDB structures. These include the monomeric conformation of an archaeal Selecase and the β-sheet fold of NusG, a single-folding RfaH homolog (Supplementary Fig. 13b). Further, all AF-cluster KaiB predictions could be reproduced successfully using this approach (Supplementary Fig. 13c).

**AF3 misassigns evolutionary restraints of dimeric XCL1 while AF2 predicts it correctly by structure memorization**

Up to now, the AF2-based methods developed to predict alternative protein conformations claim that their predictions are informed by

coevolutionary information. By our benchmarking and a previous report[15], none of them successfully predict the dimeric conformation of human XCL1, an immune system protein. Neither does AF3, which predicts an experimentally unobserved domain-swapped structure >16 Å from its experimentally observed counterpart (Fig. 5a) with average plDDTs of the ordered regions of the two top-scoring models >70. Interestingly, the coevolutionary patterns of the predicted dimer were nearly identical to those of XCL1's monomeric conformation (Fig. 5b), which all tested implementations of AF2 and AF3 capture successfully. Although the predicted dimer structures and the experimentally determined monomeric conformation differ by >14.9 Å, this result suggests that AF3 predicted the experimentally unobserved dimer by mapping some of the evolutionary couplings corresponding to intrachain (monomer) interactions to interchain (dimer) interactions instead. Unfortunately, the evolutionary couplings corresponding to monomeric lymphotactin cannot inform the prediction of dimeric lymphotactin, whose intra- and interchain interactions differ (Supplementary Fig. 14).

Though none of the AF-based methods successfully inferred dimeric XCL1's structure from evolutionary couplings, both AF2 and AF2-multimer successfully predict its structure from sequence alone (Fig. 5c). As before, both predictions were run with 0 recycles to assess whether the models associate the sequence with a structure "memorized" during training; both results suggest that AF2 predicts the dimeric form of XCL1 by sequence association rather than energy minimizing coevolutionary restraints. These results again suggest that some successful AF2 predictions are informed by structures learned during training–such as autoinhibited RfaH and the dimeric form of XCL1. Thus, we suspect that if these structures had not been in the training set, AF2 would not predict them.

## Discussion

Although AF has revolutionized protein structure prediction and protein design, its current ability to predict alternative protein conformations is limited. We tested multiple versions of AF2, the AF3 server, and two published enhanced sampling methods on 92 fold-switching proteins, which assume two distinct biologically important conformations[18,20]. The majority of both conformations of these 92 fold switchers were likely in both AF3's training set and the latest version of AF2's since they were used to train OpenFold to

match AF2's predictive performance[27]. Furthermore, our previous work shows that, when supplied with the appropriate alternatively folding template, AF2 continues to predict the dominant conformation only[16]. Here, when we combine all models from all methods and sampling techniques– >280,000 predictions in all–AF captured fold switching with a modest success rate of 35% (32/92). Furthermore, AF showed less success predicting fold switchers outside of its training set: 14% (1/7). This one success was the homolog whose fold-switching C-terminal domain had closest sequence identity to *E. coli* RfaH, a fold switcher with both conformations likely in AF's training set. Notably, AF failed to predict correct conformations of both targets outside of its training set. It missed the 3-α-helical bundle conformation of an engineered protein that switches folds in response to temperature[36]. It also predicted a conformation of the human protein isoform BCCIPα that differed completely from its experimentally determined structure[37]. Since this structure is in complex with another protein, it is possible that BCCIPα may assume the AF-predicted conformation in its uncharacterized apo state or in complex with a different binding partner. Nevertheless, AF did not predict its experimentally characterized structure. These results suggest that current implementations of AF are unlikely to foster broad discovery of yet-to-be-discovered fold switchers.

This study involved extensive sampling of fold-switching proteins, generating >500,000 structures for 99 fold switchers. Nevertheless, more sampling with more sophisticated techniques may enable AF to predict more alternative conformations not identified here. Indeed, a recent preprint indicates that another method, AFSample2[41], successfully predicts the 3-helix-bundle fold of the engineered protein Sa1. This finding is consistent with our results since a homologous sequence that assumes the helical bundle structure is in the PDB and was likely in AF's training set. Though there is value in exploring and developing these advanced sampling techniques, our results indicate that AF is currently not sensitive enough to predict many uncharacterized fold switchers from genomes using a systematic and scalable computational approach, the particular interest of our lab that inspired this study.

With the sampling and methods performed, AF has a false negative failure rate for 65% of fold switchers within its training set and 86% for fold switchers without. We estimated false positive rates by finding how many times confident predictions (with average plDDT scores ≥70) shared little structural similarity to either of the two folds. On average, the false positive rate is 43%. The variance of this rage is high: 41% of fold switchers had false-positive rates >50%, while 30% had false positive rates <3%. False positives are more problematic than false negatives because experimentally testing positive predictions is expensive and time-consuming. Therefore, these positive predictions should be interpreted with caution and cross-validated using orthogonal methods[30].

AF's inability to accurately predict and score the multiple experimentally determined conformations of fold switchers suggests that the model has more to learn about protein energy landscapes[31]. Complete understanding would enable AF to accurately predict both conformations of fold-switching proteins and their relative frequencies. AF2's lack of energetic understanding is evidenced by (1) its inability to predict ~65% of fold switchers likely in its training set, (2) its inability to predict >85% of fold switchers outside of its training set, (3) its failure to accurately score models of alternative conformations, and (4) the inability of its structure module to distinguish between low and high energy conformations. These findings are consistent with three recent reports: (1) incorrect AF2 and AF3 structure predictions of human pro-interleukin-18[42], (2) unphysical and imbalanced AF2 predictions of experimentally determined protein kinase conformations[43], and (3) incorrect AF3 predictions of the oligomeric assembly of the human lens fiber membrane intrinsic protein MP20[44]. These predictive failures lead us to conclude that AF2 and AF3 harbor little–if any–knowledge of protein thermodynamics. This conclusion is further supported by AF2's inability to predict sparsely populated states of PimA and RfaH. These findings may also apply to dynamic single-fold proteins: a study of 91 such proteins showed that AF2 was systematically unable to reproduce experimentally observed conformational diversity[45], preferentially predicting one conformation while missing the other as observed here.

This study provides evidence that AF2 has memorized certain protein conformations during training. Our results show that AF2 predicts the helical autoinhibited conformation of RfaH despite the strong coevolutionary signal that the Evoformer detects for its active β-sheet conformation. This result suggests that AF2 sometimes favors structural information learned during training over coevolutionary information detected from MSAs. Accurate predictions of several fold switchers from single sequences further supports the possibility of memorization. The Evoformer stack was not designed to predict a robust pair representation from single sequences; nevertheless, AF2 predicts protein structures within 3.0 Å of their experimentally determined structures with 0 recycles, suggesting that it has memorized their conformations. Structure memorization may sometimes drive AF3 predictions as well since it failed to predict the experimentally determined structures of BCCIPα and human pro-interleukin-18, favoring instead structures in its training set[42], though more extensive sampling with different MSA inputs would be required to test this possibility.

Some of AF's predictive unreliability appears to arise from faulty associations between sequence and structure. For instance, AF2.3.1, AF3, and AF-cluster completely miss the experimentally determined conformation of BCCIPα, instead associating its sequence with the structure of BCCIPβ, a close homolog likely in AF2's training set[37]. Further, AF2.3.1 and AF3 incorrectly predict only β-roll folds for CTDs of 4/5 fold-switching RfaH proteins with ground state α-helical conformations. AF-cluster incorrectly predicts that the β-roll CTDs of two single-folding RfaH homologs can assume α-helical conformations indicative of fold switching. Thus, unlike its recently reported performance on some KaiB proteins[15], all of which were ≥47% identical to sequences of their PDB homologs, AF-cluster does not reliably associate sequence-diverse RfaH homologs with their experimentally observed conformations.

Our results suggest a way to potentially improve AF-based predictions of fold-switching proteins. Previous work from our lab shows that coevolutionary signals for both folds of fold-switching proteins are sometimes present in MSAs[19]. Deep MSAs show strong signal for a dominant conformation, while shallower subfamily-specific MSAs show increased signal for the alternative. AF2 appears to miss this subfamily-specific information in most cases. Better results may be obtained by fine-tuning AF2 to associate MSAs of different depths with different folds, potentially strengthening the sequence-structure associations needed to predict alternative conformations of fold-switching proteins.

Deep learning models are limited by both their underlying assumptions and their training datasets. With very limited mechanistic understanding[31] and relatively few atomic resolution examples of fold switchers[18], it may not yet be possible to leverage deep learning to consistently predict this emerging phenomenon. There may be much about the protein universe–and particularly fold switching–that has not yet been observed. This dark matter is a new frontier of protein science.

## Methods

### The dataset

The dataset of fold-switching proteins having identical to high sequence similarity but assuming two distinct secondary/tertiary structures (folds) with experimentally determined structures[16] was used for the analysis (Supplementary Data 1). To determine flexibility,

we compared B-factors of fold-switching with single-folding protein regions and found no substantial difference between the two (Supplementary Fig. 15). We also analyzed normalized B-factors as follows:

$$BF_{norm} = \frac{BF - \mu_{BF}}{\sigma_{BF}}, \qquad (1)$$

where $\mu_{BF}$ is the average B-factor over a given protein structure and $\sigma_{BF}$ is its standard deviation. As in ref. 46, residues with normalized B-factors ≥ 2.0 were considered flexible; 98% of fold-switching regions had normalized B-factors <2.0, indicating that they are not particularly flexible. This analysis indicates that the experimentally determined structure of fold-switching proteins that we used accurately represent their structures rather than forcing flexible protein regions into a rigid conformation. Sequences of experimentally characterized RfaH/NusG variants[32] and two examples of folds-switching proteins identified in 2023[36,37] were also analyzed.

### Searching for fold switchers likely in AF2's training set

We wanted to determine if both pairs of PDB structures corresponding distinct conformations of the fold switchers in our dataset were likely present in AlphaFold2's training set. However, the code for AF2's complex training procedure with data (structures, sequences, and MSAs) required for the training were not made available with the official AF2 resource. The training data for OpenFold, a robust, open-source, and trainable implementation of AF2 with equivalent performance[27], have been made available, however. The set of PDB IDs and their chain information (132,000 unique chains) used to train OpenFold[12] are in a text file (duplicate_pdb_chains.txt), hosted on the Registry of Open Data on AWS (RODA) at https://registry.opendata. aws/openfold/. We matched the PDB IDs and chains in our dataset with their training set and found all.

### Defining Fold1 and Fold2

Fold-switching proteins have two distinct conformations, A and B. Proteins with higher TM-scores in the fold-switching region for at least 3 out of 5 of their AlphaFold2.3.1 predictions were designated "Fold1" and the other conformation in the protein pair was denoted as "Fold2"[16].

### AlphaFold2 (AF2) predictions

**AF2.3.1 and AlphaFold-Multimer.** The open-source version of AlphaFold2.2.0 and 2.3.1 maintained on the NIH HPS Biowulf cluster (https://hpc.nih.gov/apps/alphafold2.html) was used to generate predictions. The template database contained PDB structures and sequences released till 2022-12-31. The pipeline was run both with and without templates, the predictions from the AlphaFold2-Multimer /AF2_multimer (7) pipeline were generated using both "monomer" and "multimer" option. Supplementary Data 1b shows change in oligomeric state between the two folds/conformations.

**AlphaFold2 with single sequences.** Additional runs were performed using AF2.3.1 and AF2.2.0, with and without templates, simply putting in the target sequence in the prediction pipeline without generating an MSA, to exclude any coevolutionary information that may be present in the MSA.

### AF3 predictions

All AF3 predictions were run on the available webserver (https://golgi. sandbox.google.com). A list of copy numbers and biological molecules for each run can be found in Supplementary Data 3. Average plDDT of the XCL1 dimer was calculated using alpha carbons of residues 25–76, the folded region of the protein. Residues outside of this region had low plDDTs because they were disordered and were therefore not counted.

### Sampling prediction ensembles with AF2

**Modified implementation of SPEACH_AF.** Alanine-masked multiple sequence alignments (MSAs) were generated by identifying all amino acids in contact with a region of interest and mutating all contacting amino acids to alanine those within 4 residues of primary sequence to the region of interest[17]. The region of interest was defined as a sliding window of 11 residues that moved by increments of 1 from the beginning to the end of the fold-switching region of each of 92 proteins. Positions in the MSA corresponding to residues within 4 Å of any amino acid within a given region of interest–except those within 4 residues of primary sequence of that region–were converted to alanine except for the target sequence. Runs using the modified MSAs were carried out with AF2.2.0, with three random seeds for each MSA for a total of 15 models for each 11-residue window. In some cases, nearby windows yielded exactly the same alanine mutations to the MSA; in these cases, only one uniquely mutated MSA was preserved. A total of 77,160 predictions were generated using this method.

**AF_Cluster.** To perform more extensive sampling of conformations, AF-cluster was run with ColabFold[39] maintained on the NIH HPS Biowulf cluster (https://hpc.nih.gov/apps/colabfold.html). This module was used to generate multiple sequence alignments (MMseqs2-based routine[47]) for the proteins in our dataset using the UniProt database[48]. The AF_Cluster pipeline (https://github.com/HWaymentSteele/AF_ Cluster)[15] was then implemented to cluster the MSAs and these shallower MSAs were then used to generate predictions using Colab-Fold1.5.2 (which uses AF2.3.1) and ColabFold1.3 (utilizes AF2.2.0, to match the results presented in Wayment-Steele et al. [15]). The Colab-Fold1.3 run reproduced Wayment-Steele et al.'s predictions of both conformations of KaiB, RfaH, and Mad2. Both versions of ColabFold were run on all fold switchers, each generating 5 relaxed structures from two random seeds, 10 structures/shallow MSA, 3 recycles. Additionally, we ran ColabFold1.5.2 generating 50 relaxed models from 10 random seeds and 3 recycles on all RfaH variants not in the PDB along with Sa1 and BCCIPα. Results for these variants outside of the PDB comprise all runs. Further, we repeated the 50-structure Colab-Fold1.5.2 runs with dropout and found no increase in alternative conformation sampling.

A table of total number of predictions generated for each protocol is presented in Supplementary Data 2. All predictions following the AF_Cluster pipeline, were generated without templates, as in the original manuscript[15].

### Assessment of prediction quality

The per-residue Local Distance Difference Test (plDDT) scores (a per-residue estimate of the prediction confidence on a scale from 0 – 100), quantified by determining the fraction of predicted Cα distances that lie within their expected intervals were used to determine confident predictions[49]. The values correspond to the model's predicted scores based on the lDDT-Cα metric, a local superposition-free score to assess the atomic displacements of the residues in the model. Values ≥ 90 were denoted as high confidence, and values between 70 to 90 are deemed confident.

Predictions were compared to the original experimentally determined structures using TM-align[28], (an algorithm for sequence-independent protein structure comparisons) and root mean square deviations (RMSDs) involving backbone atoms (C, Cα, N and O) calculated using biopython's PDB.Superimposer module[50]. TM-align first generates an optimized residue-to-residue alignment based on connections among secondary structural elements using dynamic programming iterations and then builds an optimal superposition of the two structures. TM-score (ranging from 0 to 1) is reported as the measure of overall accuracy of prediction for the models after the alignment, 0.6 signifying roughly similar folds for protein regions of interest. RMSD values ≤ 5 Å were used to infer similar structures.

## Reranking predictions based on plDDT scores for AF_Cluster predictions

For an agnostic view of the pool of predictions generated for each protein, we reranked the predictions according to the percentage of confident residues (residues having plDDT scores ≥70) and then compared them to the experimental structures. The predictions were designated as Medium (≥70% residues with plDDT scores ≥ 70), Good (≥80% residues with plDDT scores ≥70) and High (≥90% residues with plDDT scores ≥70) confidence models. The predictions were rescored according to the percentage of confident residues in each pool of Medium, Good, High and All (includes all predictions for the protein) confidence models.

To determine which conformations were present among models within each of the four categories (All, Medium, Good, and High), the total number, $N_{ij}$, of models corresponding to each conformation ($i$) of each fold-switching protein ($j$) were tabulated. A given conformation ($ij$) was considered to be predicted if $N_{ij} \geq 1$.

## Defining "Complex" and "Easy" fold-switchers

Since, the 92 fold-switching protein pairs are highly heterogeneous with respect to the structural differences between conformers, a fair comparison of the predictions produced by AF2 samplings would need a separation of the targets based on their difficulty or complexity. To that end, we labelled as the fold-switching pairs as "Easy" (48) or "Complex" (45) based on RMSD between whole structures (wRMSD) and TM-scores of the fold-switching region (fsTM-score). Hence, taking both overall conformational changes and local secondary structural changes specific to the fold-switching region into account when labelling the pair. Supplementary Data 6 has all the information on the protein pair (structural changes in terms of RMSDs and TM-scores, sequence identities) including the label "Easy" or "Complex" depending on the wRMSD (>10 Å) and fsTM-score <0.5. Amyloids and fold-switchers having swapped domains (3low_A / 3mlb_F) are labelled as "Complex" too.

## Prediction success

Success rate or prediction success is defined as the fraction of proteins for which at least one prediction corresponds well (TM-score of fold-switching region>0.6[51]) to Fold1 or Fold2 (*Defining Fold1 and Fold2*). If the TM-scores for both Fold1 and Fold2 (TM-score1 and TM-score2, respectively) are greater than 0.6 the conformation is assigned to the conformation that produces the larger TM-score. The third label (other than Fold1 and Fold2) is "Other", a.k.a. experimentally unobserved predictions, are designated to those predictions with TM-scores (TM-score1 and TM-score2) less than 0.6. After reranking, we checked for prediction success in Top1 (most confident prediction overall), Top10 (10 most confident predictions overall) and All (all predictions regardless of confidence) in the pool of predictions.

## PimA intermediates

In addition to comparing the TM-scores and RMSDs between predictions and PDB structures (PDB entry 4NC9, apo state of PimA and PDB entry 2GEJ, PimA bound to GDP-Man), the intermediate states (inactive- and active-extended) were assessed by measuring the distance between the residue Arg144 that is reshuffled from an α-helix to a ß-sheet environment and Glu157 located in the α-helix on the N-terminal domain, to replicate the domain closure associated with the conformation change on going from the inactive-compact to inactive-extended, then from active-extended to active-extended states (Fig. 4. in ref. 21 explains the different conformations clearly).

## AF2Rank

We ranked high and low energy conformations as follows. Only seven fold switchers have either been found to populate two folds in solution or populate two distinct crystal forms under the same conditions[16].

These are referred to as "experimentally isoenergetic conformations" in Fig. 2b. In the remaining cases, fold-switching proteins assume a more stable "ground" state and a less stable "excited" state. Thus, we classified the remaining 86 protein pairs into "ground" and "excited-state" conformations, as previously[16]: "We define ground state in two ways: first as isolated protein when the other conformation binds a ligand, second as a preferred conformation suggested by the literature, and third as one of two bound conformers. This third definition gives AlphaFold the benefit of the doubt when both structures are ligand-bound."

Starting from this dataset[16], any proteins where one structure included only a short fragment or that had long gaps in the fold-switching region were excluded from the AF2Rank protocol. The final dataset consisted of 76 proteins (PDB IDs highlighted in supporting data, Supplementary Data 1a).

Structures corresponding to each fold-switched conformation were passed to the AF2Rank protocol[8] as templates, and the candidate structure's accuracy is assessed based on confidence scores of the AF2 output model. Before being passed to AlphaFold2, sidechain atoms were removed to prevent AF2 from using the underlying amino acid sequence to influence its prediction confidence. Beta carbons were added to glycine residues to mask their identity. AF2 was run without a MSA to remove coevolutionary influence from protein structure prediction. As in the original publication, a composite score of predicted local distance difference test (plDDT), predicted template modeling score (pTM), and template modeling (TM) score was considered to be an energy function that evaluates model quality: the more confident and closer to the experimental structure, the higher the score[8]. For each fold-switching protein, we passed AF2Rank each of the two folds as a template structure, using its amino acid sequence as the input sequence. plDDT, pTM, and composite scores were compared between the two runs to determine which fold AF2 assigns higher confidence scores. TM-scores were also calculated between the output and template structures to assess prediction quality.

To ensure that we passed the same sequence to AF2 for fold-switched conformations, we truncated extraneous N- and C-terminal residues used for purification but endogenous to their respective sequences. If one structure included a domain that was not present in the other structure, that protein was excluded from the dataset. Any short gaps in the structures were modeled with RosettaCM[52], and the top scoring Rosetta model (minimum 1000 models generated) with a TM-score ≥0.9 compared to the native structure were then selected for use. Hetero-atoms from non-standard residues such as the selenium in seleno-methionine and seleno-cysteine were replaced with their standard analogs (e.g. methionine and cysteine) using RosettaCM.

## Single sequence predictions

Single sequence predictions were performed with ColabFold1.5.5 using monomer model weights with 0 recycles, all 5 models.

## Scripts and figures

The scripts used for all analyses were written in Python3(https://www.python.org/); modules and packages used – Biopython (https://biopython.org/), pandas (https://pandas.pydata.org/), NumPy (https://numpy.org/), tmtools (https://pypi.org/project/tmtools/). PyMOL[53] (The PyMOL Molecular Graphics System, Version 2.0 Schrödinger, LLC) (https://pymol.org/2/) was used to visualize proteins, plots were created using Matplotlib[54] (https://matplotlib.org/stable/ index.html) and seaborn[55](https://seaborn.pydata.org/).

## Reporting summary

Further information on research design is available in the Nature Portfolio Reporting Summary linked to this article.

## Data availability

Data generated for the analysis, including the multiple sequence alignments, log files, example data to run scripts and to generate figures were deposited in the Zenodo database under accession code https://doi.org/10.5281/zenodo.13221957 and are also available on GitHub: https://github.com/ncbi/AF2_benchmark. The supporting data generated in this study are provided in the Supplementary Information and the Source Data file. The structural data used in this study were taken from the Protein Data Bank, details in supporting data (Supplementary Data 1) and the ones mentioned in the manuscript are listed below with their accession codes –

NMR structure of Sa1_V90T 8e6y, chain A, solution structure of PSD-1, 2fs1, chain A, Solution structure of V21C/V59C Lymphotactin/XCL1 2hdm, chain A, Crystal structure of human BCCIP beta (Native2) 7kys chain A, Crystal structure of human FAM46A-BCCIPa complex 8exf chain B, Crystal structure of the RfaH transcription factor 2oug, chain C, Crystal structure of E.coli RNA polymerase elongation complex bound with RfaH 6c6s, chain D, Wild Type Crystal Structure of Full Length Circadian Clock Protein KaiB from Thermosynechococcus elongatus BP 2qke, chain E, NMR structure of fold switch-stabilized KaiB from Thermosynechococcus elongatus, 5jyt chain A, Crystal Structure of the Mad2 Dimer 3gmh_L, chain L and 2vfx chain L. Source data are provided with this paper.

## Code availability

Code used to generate the results, analyze data and create figures for this manuscript can be found at https://github.com/ncbi/AF2_benchmark and as a Zenodo database: https://doi.org/10.5281/zenodo.13221957.

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

## Acknowledgements

We thank Carolyn Ott and Sergey Ovchinnikov for helpful discussions and Loren Looger, Marius Clore, and George Rose for commenting on the manuscript. We also thank Wolfgang Resch for technical support. This work utilized resources from the NIH HPS Biowulf cluster (http://hpc.nih.gov) and the NHLBI Biophysics Core and it was supported by the Intramural Research Program of the National Library of Medicine, National Institutes of Health (LM202011, L.L.P.).

## Author contributions

Conceptualization: L.L.P., Methodology: D.C., L.L.P., J.W.S., and E.A.C.; Software: D.C., J.W.S., L.L.P., and E.A.C, Investigation: D.C., E.A.C, J.W.S., L.L.P., J.F.T., L.A.R., M.L.; Data Curation: D.C., L.L.P.; Visualization: D.C., L.L.P., J.W.S., E.A.C., J.F.T.; Writing – original draft: D.C., L.L.P.; Writing – review & editing: D.C., L.L.P.; Supervision: L.L.P.; Project administration: L.L.P.; Funding acquisition: L.L.P.

## Funding

## Competing interests

The authors declare no competing interests.
