## [Peer Review File · Nature Communications]

AlphaFold predictions of fold-switched conformations are driven by structure memorizationReviewers' comments:

Reviewer #1 (Remarks to the Author):

After the AF2 breakthrough, scientists are now challenged with predicting native alternative conformations with precision comparable to AF2 predictions for single structures. However, current evidence remains anecdotal, and the term 'prediction' is somewhat of a euphemism, as AF2 and its various implementations (such as AF-cluster) are primarily used to generate multiple conformations, rather than identifying top-scoring conformations belonging to the native ensemble (for example apo and holo forms).

Dra. Porter's manuscript tries to address how different AF2 implementations can predict native ensemble conformations and how their scoring system allows the user to pick up correct conformations among the conformational set obtained. Authors used a dataset consisting of 93 fold-switching proteins previously characterized. Using AF2 and their different implementations to predict conformers (AF-Cluster, AF2-multimer, SPEACH_AF, etc) and structural comparisons, authors found that AF2 is able to predict multiple conformations just in ~25% of the studied proteins. Also, they found that the AF2 scoring system is unable to discriminate high/low energy conformations.

I found the manuscript interesting and technically sound. However, I am afraid that the manuscript does not provide new biological insights in the field. Overall, negative results as those shown in the manuscript are related with Dra Porter's previous publication (Protein Sci . 2022 Jun;31(6):e4353. doi: 10.1002/pro.4353). As mentioned before, the findings that AF2 could predict conformational diversity of proteins is reduced to very few biological examples and somehow those successful results are unexpected since AF2 was not designed to explore energy landscapes (see for example J Phys Chem B. 2022 Sep 1; 126(34): 6372–6383. Published online 2022 Aug 17. doi: 10.1021/acs.jpcc.2c04346). Furthermore, Dra. Porter has also raised some questions on one of those recent studies about conformational diversity prediction (bioRxiv [Preprint]. 2024 Jan 9:2024.01.05.574434. doi: 10.1101/2024.01.05.574434).

Besides these general comments, I would like to add two methodological concerns about the manuscript. I found that the set of 93 fold-switching proteins is very heterogeneous in terms of the structural differences between conformers. I have explored some of the pairs, and some of the conformers are very similar to each other (for example 1h38_D and 1qln_A) since they differ in few residues changing from beta sheet to coil, while others have indeed substantial differences. I would recommend to split the set in "easy" or "complex" cases or to correlate conformational diversity prediction capacity as a function of RMSD between conformations.

As described by the authors, several of the conformations are stabilized after protein-protein interactions with other proteins. However, some examples in the dataset are stabilized also by ions (1uxm_k and 2nam_A) that can be tricky to AF2. Other examples in the dataset are very complex examples such as the pair of conformers containing an amyloid (6vw2_A and 2kb8_A) or swapped domains (3low_A and 3m1b_F).

Finally, the use of TM-align and TM-score could be misleading when conformers (by definition with the same sequence) are compared. TM-align in its task to optimize the structural alignment between conformers can introduce gaps introducing errors in the alignment, since the alignment is already defined (same sequence). RMSD without structural optimization would be a better and reliable estimation of structural differences between conformers.

Reviewer #2 (Remarks to the Author):

In this study Chakravarty et al. introduce a detailed study on how well AF2 (as well as enhanced sampling derivatives) can reproduce and score the different available structures on fold-switching

proteins. The title of the study is quite a summary of their conclusions: AlphaFold2 has more to learn about protein energy landscapes.

While I have no doubt that this study will be interesting for a wide readership, including structural biologist, modelers, biophysicist, etc., I also believe it does not constitute a big advancement on the field, reflecting more a continuous learning. The authors themselves already concluded a couple of years ago that "AlphaFold2 fails to predict protein fold switching". While the current study is certainly more comprehensive it reaches to similar conclusions. Thus, I believe it should be more suited to a modelling specific journal.

In addition, I am a bit turned down by the comparison, when analyzing fold switching, to crystallographic structures. Switchers are less stable and more heterogeneous than single folders. They clearly do not reach the level of IDPs but one might think it twice before it accepts the existing crystals as the true folds. No one would accept such a statement for IDPs and, at least, should question it for switchers. Thus, in order to be a significant study I would expect comparison of the predicted structures with more than the mere crystals. Interpolation along the folds using biased MDs, for example, could provide a set of intermediates to compare better AG2 structures.

I am not saying that I expect AF2 to perform well on switches. In fact I believe most of the field is aware of AF2 limitations and use it basically as a structure provider, rather than a biophysical analysis tool. But a fair comparison should not be made exclusively with xtals structures, which might be significantly biased to the crystal packing in this case.

Reviewer #3 (Remarks to the Author):

Fold-switching proteins are now recognized as an important category that departs from the dominant 'one sequence-one fold' concept. While Alpha Fold 2 and other deep learning tools are correctly viewed as a breakthrough for protein structure prediction and design, they are generally capable of predicting only one native fold for a given sequence, including those with documented biologically relevant fold-switching behavior. A method (SPEACH_AF) published in 2022 that masks dominant coevolutionary patterns was suggested as a way to allow AF2 to predict alternative conformations. A very recent Nature paper from the Kern lab (PMID 37956700) reported that simply clustering input sequences for AF2 enables it to predict multiple native state structures with high confidence, an innovation with potentially enormous impact on the fields of structural biology and protein engineering.

Using a test set of 93 fold switching proteins, this manuscript from Porter and coworkers calculated a large number (>280,000) structures and found that both SPEACH_AF and AF-Cluster failed on >80% of the test set. Additionally, several alternate folds were predicted from single sequences proving that AF-cluster's predictive success in those instances did not result from coevolutionary inference. Further testing with 7 fold switching proteins outside the AF2 training set showed that neither AF2 nor AF-Cluster reliably predicted the alternate folds. The authors conclude that AF2 harbors less knowledge of protein folding energetics than has been suggested, and that when it correctly predicts the alternate structure for a fold-switching protein, this is a simple result of algorithm overtraining.

This systematic assessment of recent claims that AF2 can reliably predict multiple folds for sequences that adopt two native structures is a timely and important contribution to the field. Methods and results are presented clearly and succinctly, and I have only minor suggestions for improving the manuscript:

1. Discussion, p. 13: Please clarify what is meant by "For context, a BLAST search of all 93 sequences would have yielded 100% of both conformations."
2. The discussion seems to indicate that AF2 has a false negative failure rate of 75-85%. Can a false positive rate be estimated? Would both types of failures have the same impact on efforts to identify

new fold switching proteins in nature or design fold switching sequences?

3. It is implicit in the descriptions of fold switching that these are proteins with two distinct structures; encoding more than two stable folded states in one sequence seems thermodynamically implausible – is it? Related: how likely is AF2 to harbor any knowledge of protein folding thermodynamics? Please comment in the discussion.

Reviewer #3 (Remarks on code availability):

I examined the README file and several directories, including python scripts. I did not attempt to install or execute any code.

Reviewer #1 Comments:

After the AF2 breakthrough, scientists are now challenged with predicting native alternative conformations with precision comparable to AF2 predictions for single structures. However, current evidence remains anecdotal, and the term 'prediction' is somewhat of a euphemism, as AF2 and its various implementations (such as AF-cluster) are primarily used to generate multiple conformations, rather than identifying top-scoring conformations belonging to the native ensemble (for example apo and holo forms).

Dra. Porter's manuscript tries to address how different AF2 implementations can predict native ensemble conformations and how their scoring system allows the user to pick up correct conformations among the conformational set obtained. Authors used a dataset consisting of 93 fold-switching proteins previously characterized. Using AF2 and their different implementations to predict conformers (AF-Cluster, AF2-multimer, SPEACH_AF, etc) and structural comparisons, authors found that AF2 is able to predict multiple conformations just in ~25% of the studied proteins. Also, they found that the AF2 scoring system is unable to discriminate high/low energy conformations.

I found the manuscript interesting and technically sound. However, I am afraid that the manuscript does not provide new biological insights in the field. Overall, negative results as those shown in the manuscript are related with Dra Porter's previous publication (Protein Sci. 2022 Jun;31(6):e4353. doi: 10.1002/pro.4353). As mentioned before, the findings that AF2 could predict conformational diversity of proteins is reduced to very few biological examples and somehow those successful results are unexpected since AF2 was not designed to explore energy landscapes (see for example J Phys Chem B. 2022 Sep 1; 126(34): 6372–6383.

Published online 2022 Aug 17. doi: 10.1021/acs.jpcc.2c04346). Furthermore, Dra. Porter has also raised some questions on one of those recent studies about conformational diversity prediction (bioRxiv [Preprint]. 2024 Jan 9:2024.01.05.574434. doi: 10.1101/2024.01.05.574434).

We thank the Reviewer for their comments. In response to their concern about adding new insights to the field, we close this revised manuscript with additional results explaining why AF2 fails to predict alternative conformations and leverage this knowledge to coax AF2 to predict the alternative conformation of a fold switcher that neither the other AF2-based methods nor AF3 could capture. Specifically, we provide direct evidence that AF2 is overtrained on alternative conformations of several fold-switching proteins. This evidence strongly suggests that AF2 would not have predicted those conformations from their respective sequences, demonstrating that its predictive scope can be limited by it has "seen" in the training set. To our knowledge, these are the first results of their kind.

In further detail, we observe that AF2 predicts a protein structure distinct from the coevolutionary signal detected by its Evoformer block (p. 15, line 591 – p. 18, line 663). This observation challenges the widely accepted claim that AF2 uses coevolutionary constraints from MSAs to predict protein structure (PMID: 36563190). In fact, our result shows that it ignores or overwrites them. We also provide a several examples in which AF2 predicts the alternative conformation of proteins from single sequences with no recycles (p. 18, lines 645-663 and Figure S13). Single sequences ensure that AF2 does not leverage coevolutionary information to infer structure. No recycles ensures that AF2's structure module operates from its PDB training since the Evoformer passes it no coevolutionary information. AF2's ability to predict these conformations with such limited information indicates that its Structure module has "memorized" these conformations, a possibility that some have suggested (e.g. <https://doi.org/10.1101/2023.09.25.559256>), but to the best of our knowledge, no one has shown before. We use these observations to successfully predict the dimeric conformation of XCL1, which no other AF2-based method or AF3 predicted (p. 18, line 664-p.20, line 690).

Some of the results just mentioned (p. 18, line 664-p.20, line 690 and Figure S13) were reported in one of our *bioRxiv* manuscripts (<https://www.biorxiv.org/content/10.1101/2023.11.21.567977v2.abstract>) and are now included in this Revision (p. 18, lines 645-663 and Figure S13)—they have not been submitted for publication elsewhere. The other *bioRxiv* manuscript (10.1101/2024.01.05.574434) compares AF-cluster with Colabfold-based random sequence sampling on 3 proteins. This manuscript focuses on ~100 proteins and AF’s ability to predict minor constituents of protein energy landscapes.

The implications of this finding are important because they explain why AF2 fails to predict alternative conformations outside of its training set and delineate some of AF2’s predictive limitations. We are aware that AF2 has correctly predicted new folds from design algorithms such as RFdiffusion combined with ProteinMPNN. This is not the same as correctly associating a naturally occurring sequence with two structures. Thus, our results indicate that AF2 likely memorized some of the alternative conformations that it correctly associates with fold-switching proteins. Had it not memorized them, it would associate the sequences of fold switchers with one structure (with stronger coevolutionary signature) rather than two. Further, AF’s failure to predict the structure of BCCIP α , which may assume only one conformation, indicates that it may struggle to predict the structures of naturally occurring sequences whose training-set homologs assume different folds.

Second, we include benchmarking of AlphaFold3, released on 5/8/24, making the results in this manuscript timelier. Although we included all possible biomolecular interactions observed in experiments, AF3 successfully predicted only 7/92 fold switchers. We show that one of its predictive failures results from wrongly assigning coevolutionary restraints to inter- rather than intra-chain interactions (p. 18, line 665-p.20, line 690 and Figures 5 and S14).

Finally, we emphasize other substantial differences between our initial submission and our previous Protein Science publication. For context, the previous publication reported a straightforward application of AF2.2.0 to our set of fold-switching proteins. It also reported that AF2.2.0 predicts the conformations of fold switchers with confidences similar to single folders and higher than IDPs, which we attributed to sequence conservation (fold switchers have conservation scores similar to single folders). Based on our results, we suggested that AF2 predicts one “most probable” conformer for a given input sequence. Below is additional content not addressed by our previous publication:

- This is the first manuscript to comprehensively show that AF2 has not truly learned protein energetics. Specifically, this manuscript demonstrates that AF2 confidence metrics systematically select *against* experimentally observed protein conformations in favor of observed ones. We did not show anything like that in our Protein Science manuscript.
- While we agree that AF2 was not designed to explore energy landscapes (and cited the J. Phys. Chem B paper the Reviewer mentioned in this revision), several recent works have suggested that the model may have learned the protein folding biophysics and can provide clues to protein ensembles/dynamics (PMID: 35739160 and 36563190). Further, there have been several different examples of how manipulation of the input MSA can lead to sampling of alternate conformations with good confidence (PMID: 35994486, 37956700, and 38538622, which we cite). These papers have received lots of attention in the field, as evidenced by numerous citations and, in one case, publication in *Nature* (PMID: 37956700). Nevertheless, none of them have been systematically benchmarked on fold-switching proteins. **This is the focus of our manuscript: 95% of predictions were performed using advanced sampling methods.** Our *Protein Science* manuscript did not address these questions at all. We have included a few sentences addressing this point in the **Introduction** section, how and why this manuscript is different from our previous work (p. 4, lines 125-135):

In previous work, we hypothesized that AF2 may be using sophisticated pattern recognition to search for the most probable conformer rather than protein biophysics to model a protein's structural ensemble. This work was a straightforward implementation of an older version of AF2 (2.0) with no enhanced sampling techniques. Since then, several newly developed enhanced sampling techniques have challenged our hypothesis, proposing instead that AF2 couples coevolution with a learned energy function to predict alternative conformations. These methods were tested on a handful of targets (6-16/study), however, leaving open the questions of (1) how well they generalize across a class of proteins and (2) what systematic benchmarking results may reveal about AF2's overall ability to predict alternative protein conformations. Furthermore, AF3 has also just been released as a webserver and has not yet been tested on fold-switching proteins.

- Our earlier manuscript tested AF2's ability to predict fold switchers likely within its training set only; this one includes benchmarking of new fold switchers outside of its training set also, where its performance is considerably worse (14% success rate).
- Our previous manuscript tested only AF2.2.0 on fold-switching proteins. In addition to the methods discussed above, this new one expands that to AF2.3 and tests the multimer model and AF3, which, we find, does not improve predictions much.

Comment 1: Besides these general comments, I would like to add two methodological concerns about the manuscript. I found that the set of 93 fold-switching proteins is very heterogeneous in terms of the structural differences between conformers. I have explored some of the pairs, and some of the conformers are very similar to each other (for example 1h38_D and 1qln_A) since they differ in few residues changing from beta sheet to coil, while others have indeed substantial differences. I would recommend splitting the set in "easy" or "complex" cases or to correlate conformational diversity prediction capacity as a function of RMSD between conformations.

As described by the authors, several of the conformations are stabilized after protein-protein interactions with other proteins. However, some examples in the dataset are stabilized also by ions (1uxm_k and 2nam_A) that can be tricky to AF2. Other examples in the dataset are very complex examples such as the pair of conformers containing an amyloid (6vw2_A and 2kb8_A) or swapped domains (3low_A and 3mlb_F).

We split our 93 fold-switching pairs into "Easy" (48) and "Complex" (45) based on RMSD between whole structures (wRMSD) and TM-scores of the fold-switching region. This accounts for both the overall conformational change taking place between the pair and the secondary structural changes in the fold-switching region while deciding the intricacy of the target. The foldswitchers_label.csv file has all the information on the protein pair, including the label "Easy" or "Complex" depending on the wRMSD (>10 Å) and fsTM-score < 0.5). We labelled amyloids and the example mentioned above of swapped domains (3low_A / 3mlb_F) as "Complex" too.

After splitting the data, we re-plotted Figure 2A (prediction frequency in different categories of the predicted model, "All", "Medium", "Good" and "High") for "Easy" and "Complex" targets. The categories are defined by the quality or prediction confidence (measured either by pLDDT or pTM scores from the AF2 predictions).

The new Figures (S7 and S8) have bar plots for both "Complex" and "Easy" fold-switching pairs, at different categories of quality, ranked by confidence (% of residues predicted to have pLDDT > 70) and pTM (predicted TM generated by AF2 as a way of scoring the predictions). Interestingly, in "Top10" and "All" categories of all predicted models (ranked by confidence), ~60% of Fold1 and ~30% of Fold2 are predicted correctly for the "Complex" pairs. Whereas it is only ~40% of Fold1 and ~20% of Fold2 predicted correctly for the "Easy" pairs. Similar trend is observed for models ranked by pTM. However, as we filter predictions based on quality, "Medium", "Good" and "High", this difference is offset and lost. This could

be explained by two things - when the conformational differences are large enough (separated by larger wells in the energy landscape) AF2 may be able to detect the two alternate conformations more accurately than subtle local changes (closer to each other in the protein folding landscape). Notably, this effect is gone as we analyze “Good” or “High” quality predictions (based on AF2 internal scores, pLDDT and pTM), thus, elucidating that scoring and filtering based on the AF2 confidence metrics lead to loss of relevant information from the protein landscapes. The second reason could be related to the main structural loss function, used while training of AlphaFold2, called Frame Aligned Point Error (FAPE) that minimizes the distance between predicted and experimentally determined structures. This could help catch larger conformational differences than the more subtle ones in the “Easy” fold-switching targets. These results have been added to our Manuscript (p. 9, lines 376-378 and Table S6).

Comment 2: Finally, the use of TM-align and TM-score could be misleading when conformers (with the same sequence) are compared. TM-align in its task to optimize the structural alignment between conformers can introduce gaps introducing errors in the alignment, since the alignment is already defined (same sequence). RMSD without structural optimization would be a better and reliable estimation of structural differences between conformers.

Response 2 : As noted on p. 6, lines 249-250, we used RMSD in addition to TM-score to assess conformational differences and found similar results (Supplementary Figure 2).

Reviewer #2 (Remarks to the Author):

In this study Chakravarty et al. introduce a detailed study on how well AF2 (as well as enhanced sampling derivatives) can reproduce and score the different available structures on fold-switching proteins. The title of the study is quite a summary of their conclusions: AlphaFold2 has more to learn about protein energy landscapes.

While I have no doubt that this study will be interesting for a wide readership, including structural biologist, modelers, biophysicist, etc., I also believe it does not constitute a big advancement on the field, reflecting more a continuous learning. The authors themselves already concluded a couple of years ago that "Alphafold2 fails to predict protein fold switching". While the current study is certainly more comprehensive it reaches to similar conclusions. Thus, I believe it should be more suited to a modelling specific journal.

We thank the Reviewer for their comments. In response to their concern about adding new insights to the field, we close this revised manuscript with additional results explaining why AF2 fails to predict alternative conformations and leverage this knowledge to coax AF2 to predict the alternative conformation of a fold switcher that neither the other AF2-based methods nor AF3 could capture. Specifically, we provide direct evidence that AF2 is overtrained on alternative conformations of several fold-switching proteins. This evidence strongly suggests that AF2 would not have predicted those conformations from their respective sequences, demonstrating that its predictive scope can be limited by it has “seen” in the training set. To our knowledge, these are the first results of their kind (p. 15, line 592-p.20, line 691).

Second, we include benchmarking of AlphaFold3, released on 5/8/24, making the results in this manuscript timelier. Although we included all possible biomolecular interactions observed in experiments, AF3 successfully predicted only 7/92 fold switchers. We show that one of its predictive failures results from wrongly assigning coevolutionary restraints to inter- rather than intra-chain interactions (p. 18, line 665-p.20, line 690 and Figures 5 and S14).

Further, the purpose of our manuscript is to ground the field by showing the limitations of AF2 very clearly and concretely. AF2 has not learned protein energetics. Consequently, it cannot predict alternative

conformations outside of its training set, and it often misses those within its training set. Similar grounding papers, such as Terwilliger, T.C., et al. “AlphaFold predictions are valuable hypotheses and accelerate but do not replace experimental structure determination”, have been published in other Nature Journals (Nat Methods 21, 110–116 (2024). <https://doi.org/10.1038/s41592-023-02087-4>). These papers can make impactful contributions. Furthermore, the importance of our manuscript is highlighted by a recent citation in a *Cell* commentary (Miller, Edward B., et al. "Enabling structure-based drug discovery utilizing predicted models." *Cell* 187.3 (2024): 521-525. <https://doi.org/10.1016/j.cell.2023.12.034>). The authors chose to cite the preprint of this work instead of the already published work in *Protein Science*, underscoring the novelty of this work. We agree with the reviewer’s opinion that this work “will be interesting for a wide readership, including structural biologist, modelers, biophysicist”, corroborated by the preprint being cited six times already. This is why we feel that *Nature Communications* is an appropriate place to publish it.

Comment: In addition, I am a bit turn down by the comparison, when analyzing fold switching, to crystallographic structures. Switchers are less stable and more heterogeneous than single folders. They clearly do not reach the level of IDPs but one might think it twice before it accepts the existing crystals as the true folds. No one would accept such a statement for IDPs and, at least, should question it for switchers. Thus, in order to be a significant study, I would expect comparison of the predicted structures with more than the mere crystals. Interpolation along the folds using biased MDs, for example, could provide a set of intermediates to compare better AF2 structures.

Our original set of fold switchers was carefully curated to remove alternative conformations resulting from crystal contacts. The biological relevance of both conformations of our fold switchers was part of the basis for its publication in PNAS (PMID: 29784778). As stated in that publication, “To confirm biological relevance, we reviewed the literature reporting both structures in the pair and required that it (*i*) claims that the switch is biologically relevant and (*ii*) reports the trigger for the switch. This step also eliminates false positives resulting from weak electron density, crystal packing artifacts, insufficient data-derived constraints, and controversial structures.” The quality of this curation has stood the test of time: we were able to observe coevolutionary signatures unique to both conformations of many proteins from this set (see paper Schafer, J.W., Porter, L.L PMID: 37673981). Coevolutionary signatures corresponding to the two conformations are hardly artefactual and would not result from crystal contacts alone. We also note that 25% of the structures in this set of proteins were not determined by X-ray crystallography but were rather solved by NMR or cryo-EM. We omitted one fold switch pair in this Revision so that both conformations of all pairs were represented in OpenFold’s training set. Further, all targets outside of AF2’s training set were not solved by X-ray crystallography.

We also took the Reviewer’s advice and compared AF2 predictions with two experimentally characterized fold-switching intermediates: PimA described by NMR and ensembles generated by steered-MD (PMID: 32434931) and RfaH, whose sparsely populated intermediate was characterized by NMR experiments (PMID: 36255050). To our knowledge, these are the only two characterized fold-switching intermediates with experimental support. AF2 (AF2.3.1 with/without templates, AF_cluster using both versions of Colabfold 1.3 and 1.5.2), did not predict any of the intermediates described in these papers, again demonstrating AF2’s limitations in producing an ensemble of protein conformations. These results are consistent with a recently released preprint comparing AF2 predictions and steered MD trajectories of 4 other proteins (<https://www.biorxiv.org/content/10.1101/2024.04.16.589792v1.abstract>). In the authors’ words: “significant minima of free energy surfaces remain undetected.” These findings are reported on p.10, line 432-p.11, line 479.

Comment: I am not saying that I expect AF2 to perform well on switches. In fact, I believe most of the field is aware of AF2 limitations and use it basically as a structure provider, rather than a biophysical

analysis tool. But a fair comparison should not be made exclusively with xtals structures, which might be significantly biased to the crystal packing in this case.

As mentioned before, our test set was curated to remove structures due to crystal packing, and ~25% the structures in our set were solved by other methods, such as NMR and cryo-EM. In fact, none of the conformations of the proteins outside AF2's training were solved by crystallography. Rather, NMR and cryo-EM were used to three-dimensional structures and assess secondary structures.

Furthermore, from what we can see, it seems that much of the field is not aware of AF2's limitations. In fact, both peer-reviewed papers presenting the MSA subsampling methods benchmarked in our submission called the findings of our Protein Science paper into question. Both suggested that better MSA sampling methods should yield successful predictions of metamorphic proteins. **While our Protein Science paper does not explore MSA subsampling methods at all, 95% of the predictions in our present work come from MSA subsampling.** We find that MSA subsampling hardly makes a difference. Nevertheless, both MSA subsampling methods have been highly cited in the field—again indicating many people believe AF2 should be able to predict fold switching. In further detail:

1. The Nature paper that recently reported the AF-cluster method benchmarked in our manuscript clearly assumes the opposite (PMID: 37956700). In their introduction, the manuscript states, “We demonstrate that a simple MSA subsampling method—clustering sequences by sequence similarity—enables AF2 to predict both states of the metamorphic proteins...”. Although the authors show it to be true in 8 targets from 3 protein families, our work clearly shows that this method does not generalize. Furthermore, it helps to explain why: AF2 is unlikely to generate reliable predictions of alternative conformations unless they were in its training set.
2. Additionally, the authors of the SPEACH_AF method state in their discussion: “That alternate conformations are encoded in the MSA and the ability of the method to yield multiple secondary structural elements (Fig A in S1 Text) would support this methodology in examining fold-switch proteins where the standard AF2 pipeline was generally unable to model both conformations [20]”. Our Protein Science paper is reference 20. As you can see, they thought that smarter MSA sampling should lead to successful predictions of metamorphic proteins, but it usually doesn't.
3. Most AF2 users rely on confidence (pLDDT) scores to select reliable predictions. Our work shows systematically that the pLDDT scores are not always reliable. Instead, they select against experimentally consistent conformations in favor of experimentally inconsistent ones for fold-switching proteins at least.

Reviewer #3 (Remarks to the Author):

Fold-switching proteins are now recognized as an important category that departs from the dominant ‘one sequence-one fold’ concept. While Alpha Fold 2 and other deep learning tools are correctly viewed as a breakthrough for protein structure prediction and design, they are generally capable of predicting only one native fold for a given sequence, including those with documented biologically relevant fold-switching behavior. A method (SPEACH_AF) published in 2022 that masks dominant coevolutionary patterns was suggested as a way to allow AF2 to predict alternative conformations. A very recent Nature paper from the Kern lab (PMID 37956700) reported that simply clustering input sequences for AF2 enables it to predict multiple native state structures with high confidence, an innovation with potentially enormous impact on the fields of structural biology and protein engineering.

Using a test set of 93 fold switching proteins, this manuscript from Porter and coworkers calculated a large number (>280,000) structures and found that both SPEACH_AF and AF-Cluster failed on >80% of the test set. Additionally, several alternate folds were predicted from single sequences proving that AF-cluster's predictive success in those instances did not result from coevolutionary inference. Further testing with 7 fold switching proteins outside the AF2 training set showed that neither AF2 nor AF-Cluster

reliably predicted the alternate folds. The authors conclude that AF2 harbors less knowledge of protein folding energetics than has been suggested, and that when it correctly predicts the alternate structure for a fold-switching protein, this is a simple result of algorithm overtraining.

We thank the Reviewer for their positive comments.

This systematic assessment of recent claims that AF2 can reliably predict multiple folds for sequences that adopt two native structures is a timely and important contribution to the field. Methods and results are presented clearly and succinctly, and I have only minor suggestions for improving the manuscript:

1. Discussion, p. 13: Please clarify what is meant by “For context, a BLAST search of all 93 sequences would have yielded 100% of both conformations.”

The protein pairs in our dataset share high sequence similarity (mean of 99%), so if one were to take the primary sequence of a fold-switching protein from the set and perform a BLAST search against sequences of solved structures, the search would hit both PDB structures for the pair. Thus, performing better than all possible implementations AF2, that catches a modest ~25% of both conformations correctly. This has been clarified in the Discussion (p. 20, lines 704-707).

2. The discussion seems to indicate that AF2 has false negative failure rate of 75-85%. Can a false positive rate be estimated? Would both types of failures have the same impact on efforts to identify new fold switching proteins in nature or design fold switching sequences?

We estimated false positive rates by finding how many times confident predictions (with average pLDDT scores greater than 70) shared no significant structural similarity to either of the two folds (TM-scores of fold-switching regions < 0.5). On average the false positive rate is 43%, notably, the standard deviation is high as well. In details, there are roughly 41% of fold-switchers with false positives greater than 50%, however, there are also 30% of them with a false positive rate of less than 3%. These numbers indicate that for some fold-switchers, AF2 can produce more accurate and confident predictions, however for many of them the confident predictions do not resemble any experimentally consistent structures. These estimates are now in the Discussion (p. 21, lines 739-747).

Further, we discussed an example of an AF2-multimer prediction of RfaH, a hybrid α -helical/ β -sheet fold with high confidence for its fold-switching C-terminal domain (**Figure S6**), this structure is not consistent with experiment. This unbiased nature of the false positive rate makes it difficult to discern correct predictions of alternate conformations from unphysical ones.

3. It is implicit in the descriptions of fold switching that these are proteins with two distinct structures; encoding more than two stable folded states in one sequence seems thermodynamically implausible – is it? Related: how likely is AF2 to harbor any knowledge of protein folding thermodynamics? Please comment in the discussion.

Fold-switching proteins have reasonably deep multi-funnels with each low energy basin representing a distinct well folded conformation; interesting point about fold-switching proteins is that *stability counterbalances the phenomenon of fold switching*, i.e., with decrease in stability alternate folds otherwise inaccessible become feasible (Kulkarni P, *et. al.* “Structural metamorphism and polymorphism in proteins on the brink of thermodynamic stability”. *Protein Sci.* 2018;(9):1557-1567.). One can imagine these folds to be “metastable”, i.e., the energy landscapes of fold-switchers may have two or more wells/basins that are usually separated by a free energy barrier of ≤ -3 Kcal/mol (comparatively, $\Delta G_{\text{folding}}$ ranges from -15 to -5 kcal/mol for most globular single folding proteins) (Dishman AF, Volkman BF. “Design and discovery

of metamorphic proteins”. *Curr Opin Struct Bio.* 2022; 74:102380.). Although three+-state fold switchers have not been discovered, they may exist in Nature. Time will tell.

Based on our results, AF2 harbors little—if any—knowledge of protein folding thermodynamics. In other words, its energy function cannot reliably predict relative populations of highly and sparsely populated states. This has been added to the Discussion (p. 21, lines 758-760).

REVIEWER COMMENTS

Reviewer #1 (Remarks to the Author):

First of all I would like to thank Dra. Porter for answering most of my previous concerns. I find this version significantly more appealing than the previous one, due to the inclusion and development of new information and concepts by Dra. Porter and her team. However, the new version is somehow difficult to read, possibly for the long extension and/or an absence of a clear "takeaway" message.

As mentioned in the manuscript, fold-switching protein conformers are part of the native ensemble for a given protein. As was previously shown by different manuscripts, AF is unable to predict conformational diversity in fold-switching proteins and in "regular" proteins without containing a "fold-switch". Then, for me the most important part of the paper is the "why is AF unable to predict different conformations..." contained in the sections devoted to "memorization", where, with clever protocols, Dra. Porter shows that AlphaFold is unable to predict alternative structures according to coevolutionary information due to the bias during the learning process. I think that the manuscript could be reformulated to showcase these new and interesting results. Hope the following comments help to do that.

I don't understand the overall discussion about "energetics" in the manuscript. I think that classification of "high" and "low" energy conformations is missing in the manuscript. How do the authors select that a given conformation coming from fold-switching proteins is "high" or "low"? I do understand that most populated conformation in a given equilibrium is most of the time the conformation with lower energy. This is mostly straight forward for apo-holo forms where apo forms are in general the most abundant and the one with the lowest energy.

Same as above, from the manuscript's title "AlphaFold has more to learn about protein energy landscapes" as several papers pointed out before, AF did not resolved the protein folding problem (as it is also commented in the Discussion section of the manuscript), so I should say that AF has to learn "all" about folding (energy landscapes). As far as I understand, AF was indeed not developed to deal with solving the protein folding problem! So, in some way I think that the title of the manuscript does not represent the overall work in the manuscript. We know that AF is unable to predict conformational diversity (as it hasn't resolved the protein folding problem) so, the main contribution of this manuscript should focus on the "why" question, let's say "memorization".

Sentence in line 717 "AF2 and AF3 were trained mainly on single-fold proteins". However, the issue seems to be not that they were trained with single-fold proteins, but rather with single conformations of proteins in general. But if AF would have been trained with a redundant collection of conformations for each protein, the "memorization" would help to predict conformational diversity? What would be the main differences in terms of AF prediction between a single-fold protein with a conformational diversity of, let's say, 10Å and a "fold-switch protein" with a conformational diversity of 3Å?

The conclusions in lines 751-758 are very similar to those regarding conformational diversity for "single-fold proteins" in the 2022 paper "Impact of protein conformational diversity on AlphaFold predictions." if fold-switch proteins are a special case of the conformational diversity of "single-fold proteins," can the conclusions of your paper be extended to the latter?

Lines 659-664, I think this conclusion is not well supported in the sense BLAST do not recover "conformations", recover sequences

Reviewer #2 (Remarks to the Author):

I admit that the current paper is significantly better than the previous one. For that I congratulate the authors.

Still I see this paper as a continuation of previous results, adding on top of a significant amount of current data on AF2 already published, and just after AF3 came out. Surely one year ago this paper belonged here. Today I am not seeing it.

And I reiterate that the comparison to Xtal structures does not seem fair to me. It is not that we are going to be searching for direct crystal contacts in the xtal, but surely highly disorder/flexible regions might be significantly different in the xtal overall packing environment than in solution. I need a more systematic assessment that xtals are correct in this context.

Reviewer #3 (Remarks to the Author):

Chakravarty and coauthors have revised and expanded this manuscript with important clarifications and new results (particularly Fig 5 and S14) that bolster their conclusions substantially. Their responses to the other reviewers have further convinced me of the importance of this study: I concur with the authors' assertion that the vast community of Alpha Fold users by and large assumes that AF2 predictions reflect some physical knowledge of protein folding. Naturally, those users would readily accept the suggestion, most notably by Wayment-Steele (PMID 37956700), that AF2 can infer structural contacts from the evolutionary information contained in clustered multiple sequence alignments. Instead, AF2 is a very efficient pattern recognition machine.

Reviewer #3 (Remarks on code availability):

I did not install and run the code, but I browsed all folders in the github repository and downloaded and reviewed the .ppt files.

First of all I would like to thank Dra. Porter for answering most of my previous concerns.

I find this version significantly more appealing than the previous one, due to the inclusion and development of new information and concepts by Dra. Porter and her team. However, the new version is somehow difficult to read, possibly for the long extension and/or an absence of a clear “takeaway” message.

As mentioned in the manuscript, fold-switching protein conformers are part of the native ensemble for a given protein. As was previously shown by different manuscripts, AF is unable to predict conformational diversity in fold-switching proteins and in “regular” proteins without containing a “fold-switch”. Then, for me the most important part of the paper is the “why is AF unable to predict different conformations...” contained in the sections devoted to “memorization”, where, with clever protocols, Dra. Porter shows that Alphafold is unable to predict alternative structures according to coevolutionary information due to the bias during the learning process. I think that the manuscript could be reformulated to showcase these new and interesting results. Hope the following comments help to do that.

We thank the Reviewer for their positive comments and address their constructive criticism below.

I don't understand the overall discussion about “energetics” in the manuscript. I think that classification of “high” and “low” energy conformations is missing in the manuscript. How do the authors select that a given conformation coming from fold-switching proteins is “high” or “low”? I do understand that most populated conformation in a given equilibrium is most of the time the conformation with lower energy. This is mostly straight forward for apo-holo forms where apo forms are in general the most abundant and the one with the lowest energy.

We thank the Reviewer for mentioning this. Most fold-switching events are triggered by ligand binding, so the Reviewer's comment about apo-holo is exactly right. Because we had published this dataset previously (PMID: 35634782), we had not mentioned it in our previous submission. We now mention how we identified high- and low-energy conformers in Methods (p. 28, lines 746-755):

*We ranked high and low energy conformations as follows. Only seven fold switchers have either been found to populate two folds in solution or populate two distinct crystal forms under the same conditions¹⁶. These are referred to as “experimentally isoenergetic conformations” in **Figure 2B**.*

In the remaining cases, fold-switching proteins assume a more stable “ground” state and a less stable “excited” state. Thus, we classified the remaining 86 protein pairs into “ground” and “excited-state” conformations, as previously¹⁶: “We define ground state in two ways: first as isolated protein when the other conformation binds a ligand, second as a preferred conformation suggested by the literature, and third as one of two bound conformers. This third definition gives AlphaFold the benefit of the doubt when both structures are ligand-bound.”

Same as above, from the manuscript’s title “AlphaFold has more to learn about protein energy landscapes” as several papers pointed out before, AF did not resolved the protein folding problem (as it is also commented in the Discussion section of the manuscript), so I should say that AF has to learn “all” about folding (energy landscapes). As far as I understand, AF was indeed not developed to deal with solving the protein folding problem! So, in some way I think that the title of the manuscript does not represent the overall work in the manuscript. We know that AF is unable to predict conformational diversity (as it hasn’t resolved the protein folding problem) so, the main contribution of this manuscript should focus on the “why” question, let’s say “memorization”.

Because of this comment, we have revised the title to **AlphaFold predictions of fold-switched conformations are driven by structure memorization, not learned folding energetics**

We revised the Abstract to tell a more cohesive story also.

Sentence in line 717 “AF2 and AF3 were trained mainly on single-fold proteins”. However, the issue seems to be not that they were trained with single-fold proteins, but rather with single conformations of proteins in general. But if AF would have been trained with a redundant collection of conformations for each protein, the “memorization” would help to predict conformational diversity?

What would be the main differences in terms of AF prediction between a single-fold protein with a conformational diversity of, let's say, 10Å and a "fold-switch protein" with a conformational diversity of 3Å?

We thank the Reviewer for pointing this out. We deleted those last two sentences because a much larger analysis is required to answer the Reviewer’s question about conformationally-diverse single folders versus fold switchers. This analysis is beyond the scope of the current work, though we

hope to provide clear answers soon. However, we add some speculation to the Discussion. See our response to the next point.

The conclusions in lines 751-758 are very similar to those regarding conformational diversity for “single-fold proteins” in the 2022 paper “Impact of protein conformational diversity on AlphaFold predictions.” if fold-switch proteins are a special case of the conformational diversity of “single-fold proteins,” can the conclusions of your paper be extended to the latter?

Based on the draft we submitted, lines 751-758 correspond to references, leaving us a little unsure of the exact part of the manuscript to which the Reviewer refers. However, we cited the reference and suggested that our results may extend to single-fold proteins in lines 545-548:

These findings may also apply to dynamic single-fold proteins: a study of 91 such proteins showed that AF2 was systematically unable to reproduce experimentally observed conformational diversity⁴⁴, preferentially predicting one conformation while missing the other as observed here.

Lines 659-664, I think this conclusion is not well supported in the sense BLAST do not recover “conformations”, recover sequences

BLAST can recover conformations when it is used to search the PDB. Indeed, we have used it for this previously (PMIDs: 29784778 and 37264049). However, to avoid confusion, we removed lines 659-664 from the Discussion.

Reviewer #2 (Remarks to the Author):

I admit that the current paper is significantly better than the previous one. For that I congratulate the authors.

We thank the Reviewer for their positive comment.

Still I see this paper as a continuation of previous results, adding on top of a significant amount of current data on AF2 already published, and just after AF3 came out. Surely one year ago this paper belonged here. Today I am not seeing it.

We respectfully disagree with the Reviewer on this point. Our manuscript includes benchmarking of methods published in 2024 (AF-cluster and AF3). Therefore, it could not have been published one year ago.

And I reiterate that the comparison to Xtal structures does not seem fair to me. It is not that we are going to be searching for direct crystal contacts in the xtal, but surely highly disorder/flexible regions might be significantly different in the xtal overall packing environment than in solution. I need a more systematic assessment that xtals are correct in this context.

In response to this comment, we compared the B-factors of the fold-switching regions of proteins to the single-folding regions. If the fold-switching regions were more flexible, we would expect to see significantly higher B-factors. We don't: the distributions are nearly identical. Furthermore, while residues with normalized B-factors ≥ 2.0 are typically considered "flexible" (PMID: 14691223), 98% of the residues in fold-switching proteins have normalized B-factors < 2.0 , indicating that they are not notably flexible. We have included these results in Figure S15 and reported them in lines 607-616.

Further, numerous publications show excellent agreement between observed and calculated residual dipolar couplings (RDCs, solution measurements of internuclear vector orientations) and ordered regions of crystal structures (PMIDs: 16343537, 37448874, 34757725, 37330294, 34766756). Consistent with this view, both PimA structures in our manuscript were solved by crystallography and corroborated by solution NMR experiments and MD simulations. Indeed, crystal structures typically agree better with RDCs than NMR structures refined without them (PMID: 15139819). Thus, unless strong contrary evidence is provided, ordered regions of crystal structures—such as those in our dataset—are expected to match solution structures.

Reviewer #3 (Remarks to the Author):

Chakravarty and coauthors have revised and expanded this manuscript with important clarifications and new results (particularly Fig 5 and S14) that bolster their conclusions substantially. Their responses to the other reviewers have further convinced me of the importance of this study: I concur with the authors' assertion that the vast community of Alpha Fold users by and large assumes that AF2 predictions reflect some physical knowledge of protein folding. Naturally, those users would readily accept the suggestion, most notably by Wayment-Steele (PMID 37956700), that AF2 can infer structural contacts from the evolutionary information contained in clustered multiple sequence alignments. Instead, AF2 is a very efficient pattern recognition machine.

We thank the Reviewer for their positive comments and hope that our manuscript can set the record straight soon.

Reviewer #3 (Remarks on code availability):

I did not install and run the code, but I browsed all folders in the github repository and downloaded and reviewed the .ppt files.

REVIEWERS' COMMENTS

Reviewer #1 (Remarks to the Author):

Dra. Porter has addressed all my concerns.

Reviewer #2 (Remarks to the Author):

I will not oppose to publication. But in my opinion this manuscript might not belong here. Do not get me wrong. It is a very good manuscript, no doubt. With great execution. But lacks originality and novelty. It is too incremental with previous work by the same authors.

The fact that they justify one of my questions with adding just new benchmark data not available one year ago clearly states it.